

# How do geological map details influence geology-streamflow relationships in large-sample hydrology studies?

Thiago V. M. do Nascimento[1,3,] Julia Rudlang[2], Sebastian Gnann[4], Jan Seibert[3], Markus Hrachowitz[2] and Fabrizio Fenicia[1]

[1]Eawag: Swiss Federal Institute of Aquatic Science and Technology, Dübendorf, Switzerland.

[2]Department of Water Management, Faculty of Civil Engineering and Geosciences, Delft University of Technology, Delft, Netherlands.

[3]Department of Geography, University of Zurich, Zurich, Switzerland

[4]Chair of Hydrology, Faculty of Environment and Natural Resources, University of Freiburg, Germany

*Correspondence to*: Thiago V. M. do Nascimento (thiago.nascimento@eawag.ch)

**Abstract.** Large-sample hydrology datasets have advanced hydrological research, yet the impact of landscape attribute level of detail on inferring dominant streamflow generation processes across scales remains underexplored. This study investigates the role of geology using maps of increasing detail—global, continental, and regional—each reclassified into four relative permeability classes. These geological attributes, combined with topography, soil, vegetation, land use and climate attributes, were analyzed across 4,000 European catchments from the EStreams dataset, to identify dominant controls on streamflow signatures. We conducted analyses at three scales: large (63 European river basins), intermediate (the Moselle basin), and small (five Moselle sub-catchments). The large-scale study used global and continental maps, while the intermediate and small-scale experiments also incorporated regional maps. On the large scale, no consistent correlation emerged between baseflow and landscape attributes, though landscape effects outweighed climate influences. The continental map generally showed stronger correlations than the global map, but with tradeoffs in the number of geological classes versus spatial resolution. At the intermediate scale, geology transitioned from being insignificant to dominant as map detail increased, underscoring the importance of refined geological data. The small-scale experiment mirrored large-scale findings, showing varying dominant controls across catchments. However, the regional map provided consistent, physically meaningful correlations, aligning with established hydrological understanding. Overall, this illustrates the considerable benefit of integrating detailed, region-specific geological data into large sample hydrology studies. Overall, our findings have implications for hydrological regionalization and the prediction of streamflow in ungauged catchments.

## 1. Introduction

The availability of large-sample hydrology (LSH) datasets, which include hydrometeorological time series and catchment attributes for hundreds to thousands of catchments, has grown significantly in recent years (Addor et al., 2017; Chagas et al., 2020; Coxon et al., 2020; Fowler et al., 2021; GRDC, 2024; Helgason and Nijssen, 2024; Höge et al., 2023; Klingler et al., 2021; Kratzert et al., 2022; do Nascimento et al., 2024a). This expansion enabled studies across a wide range of catchments worldwide (Addor et al., 2018; Almagro et al., 2024; Beck et al., 2015; Ibrahim et al., 2024; Kratzert et al., 2019; Kuentz et al., 2017; Nearing et al., 2024; van Oorschot et al., 2024; Wu et al., 2021).

A key objective of most of these studies has been to investigate which climate or landscape attributes predominantly influence specific aspects of streamflow response. While from prior hydrological understanding one would expect that the streamflow response is driven by both climate and landscape characteristics (Bloomfield et al., 2021; Gnann et al., 2021; do Nascimento



et al., 2024b), many LSH studies highlighted climate as the dominant or sole driver of streamflow response (Addor et al., 2018; Beck et al., 2015; Huang et al., 2021; Kuentz et al., 2017; Wu et al., 2021).

This raises an important question: why does the influence of landscape appear weak or even absent in large-sample studies? One possibility is that the inherent uniqueness of individual catchments prevents the formulation of generalized relationships that hold true across extensive regions. Another is that the landscape indicators extracted from landscape maps and employed in these studies lack sufficient information content. Finally, the level of detail in these maps may be insufficient, obscuring landscape influences—such as when diverse and hydrologically relevant landscape features are grouped into too few classes.

The concept of uniqueness of place in hydrology, postulated by Beven (2000), describes how specific combinations of climate, geology, topography, vegetation, and human influences shape hydrological processes within individual landscapes. This inherent uniqueness complicates the regionalization of dominant processes and their relationships with catchment landscape attributes, as assumptions valid in one context may not apply in another. For instance, variability in the baseflow index can invariably be related to geology (Fenicia and McDonnell, 2022; Pfister et al., 2017), climate (Addor et al., 2018; Beck et al., 2015; Mwakalila et al., 2002), topography (Santhi et al., 2008), soils (Schneider et al., 2007) or land use (Zomlot et al., 2015) in different regions. As a result, efforts to develop generally valid and applicable relationships to predict streamflow signatures across large regions are often hindered by the complex and localized interactions between landscape and hydrological dynamics.

The information content of summary statistics used to capture landscape and streamflow time series is also an important aspect in hydrology. While summary indicators distil complex data into manageable metrics, they often fail to capture the full depth of information present in spatial maps or temporal patterns. For example, the spatial distribution or the temporal variability of landscape attributes is typically lost when exclusively relying on aggregated metrics (Floriancic et al., 2022; Tarasova et al., 2024). Additionally, key details about landscape characteristics may not be transferred into readily available indicators. For example, the same rock types can exhibit wide varying properties depending on factors such as the degree of weathering, extent of fracturing, presence of secondary porosity and geological ages (Gnann et al., 2021). Such characteristics are often absent from standard variables readily available in LSH datasets. Fenicia and McDonnell (2022) demonstrate that correlations between streamflow and baseflow index become apparent only after developing tailored indicators, such as classifying geological units by their permeability. This highlights the necessity of designing indicators that are sensitive to the underlying processes and heterogeneity of the landscape.

Landscape maps also vary in spatial resolution and the level of detail they provide, which affects their usefulness in hydrological studies. Several LSH studies suggested that the weak correlations between landscape attributes and streamflow can be attributed to the level of detail of global landscape maps (Addor et al., 2018; Beck et al., 2015; Kratzert et al., 2019). In general, global maps tend to be coarser and less accurate, while regional maps offer greater detail and precision. As Addor et al. (2020) noted, readily available standardized datasets like global maps enable more objective comparisons of catchments from different locations, and global maps are thus often used in large-sample studies. However, this consistency typically comes at the cost of detail, reducing the accuracy of the represented landscape heterogeneity in such global maps. For example, the Global Lithological Map (GLiM) (Hartmann et al., 2012) groups rock types globally into 16 classes at their first level. While the classification facilitates evaluation by categorizing different rock types, there is, for instance, only one class for siliciclastic rocks, which can encompass very distinct rock types like sandstones and shales with vastly different permeabilities. Insufficient detail in geological data can then lead to inaccuracies in predicting infiltration, groundwater flow, and storage dynamics using models (Blöschl and Sivapalan, 1995).

Considerable attention has been devoted to how the uncertainty of climate data affects LSH study outcomes. Clerc-Schwarzenbach et al. (2024) found that regional meteorological forcing variables available in CAMELS datasets (Addor et al., 2017; Chagas et al., 2020; Coxon et al., 2020) provided more accurate and realistic inputs for hydrological models than



the global product ERA5 (Hersbach et al., 2020), which was used to derive meteorological forcings for catchments worldwide in the Caravan dataset (Kratzert et al., 2022).

In contrast, far less attention has been given to how the quality of landscape data affects our understanding of dominant hydrological processes in LSH studies. There remains a lack of systematic studies applying a standardized dataset to investigate how correlations between streamflow signatures and geological attributes differ with varying map detail level, from larger to smaller scales. Consequently, it is challenging to discern whether the variability in findings about dominant controls across different regions genuinely reflects the uniqueness of place or whether it arises from inconsistencies in map quality and

methodological approaches across studies. Addressing this gap is crucial for improving the reliability of insights derived from LSH analyses and advancing our ability to regionalize hydrological processes effectively.

In this study, we utilized the EStreams dataset (do Nascimento et al., 2024a), which covers hydro-meteorological and landscape attributes for thousands of catchments over pan-European territory. Our analysis focused on streamflow signatures, alongside climatic and landscape attributes derived from this dataset. Additionally, we incorporated two geological maps at continental

(Duscher et al., 2019; Günther and Duscher, 2019) and regional scales (AGE, 2024; BDLISA database, 2024; GÜK200, 2024), complementing the global geology map in the original dataset. These maps offer a higher level of detail, enhancing the representation of geological attributes.

Here we conducted a multi-scale analysis: (a) a large-scale assessment of 63 river basins across Europe, containing a total of 4,469 sub-catchments, (b) an intermediate-scale analysis of one river basin – the Moselle – with 152 sub-catchments —selected

for its numerous previous studies and the availability of a regional-scale geology map, and (c) a small-scale investigation of five catchments (121 sub-catchments therein) within the Moselle. At each scale, each study basin contained a variable number of nested (sub)catchments enabling an analysis of influence factors on the spatial variability of streamflow signatures. This stepwise approach balances depth with breadth (Gupta et al., 2014), conducting progressively more detailed analyses from large to small scales. The key objective is to describe and quantify the role of geological map detail in explaining correlation

patterns with hydrological signatures to improve hydrological interpretation. This stepwise approach enables us to address the core question: How does the level of detail in landscape maps impact our understanding and our ability to quantitatively describe dominant streamflow generation processes?

Accordingly, this study aims to:

- **Large-scale (63 river basins)**: Quantify the relative influences of landscape and climate attributes on streamflow
signatures across 63 European basins. Identify general pattern in dominant controls on streamflow signatures and evaluate how detail in geological information from global and continental maps affects this interpretation, in particular with respect to baseflow-related signatures.
- **Intermediate-scale (Moselle basin)**: Analyze dominant controls on streamflow signatures within the Moselle basin, using global, continental, and regional geological maps. Investigate how increasing geological detail influences the
interpretation of basin behavior.
- **Small-scale (5 catchments in the Moselle basin)**: Examine dominant controls on streamflow signatures across five catchments in the Moselle. Assess the consistency of emerging patterns in terms of dominant controls on streamflow signatures and evaluate how different levels of geological detail influence correlations and inferences on baseflow generation processes.

Our hypotheses of the effect of underlying geological maps on the correlation between geological attributes and hydrological signatures were as follows:





- Increased correlation with higher map detail: More detailed geological maps are expected to enhance the correlation between derived geology attributes and streamflow signatures, assuming that geology influences the streamflow regime. It is important to note that all the three maps (global, continental and regional) are vector-based, rather than raster-based, meaning they are not inherently tied to a spatial resolution. However, it is reasonable to assume that the level of detail increases with the scale of the map.
- Consistency of correlations with physical understanding: As map detail increases, we expect the correlations to become progressively more consistent with physical understanding. For example, the baseflow index ($\sigma_{BFI}$) should show positive correlations with high-permeability geological attributes, as high bedrock permeability favors groundwater flow, which facilitates baseflow (Bloomfield et al., 2021; Fenicia and McDonnell, 2022). We also anticipate that, because of this physical consistency, the correlations will become less variable across different regions.

This paper is structured as follows. Section 2 describes the data and the study area. Section 3 describes the methodology applied. Section 4 presents the main results and is divided into three parts: large-scale analysis (4.1), the intermediate-scale (Moselle basin) (4.2) and the small-scale (4.3). Section 5 discusses the results, mainly according to the key hypothesis introduced in this section. Finally, section 6 summarizes the main conclusions.

## 2. Data

### 2.1. The EStreams dataset

The data used in this study is obtained from the EStreams dataset (do Nascimento et al., 2024a). EStreams provides catchment delineations, meteorological time-series, hydro-climatic signatures, landscape attributes (topography, soils, geology, vegetation, and land use) for over 17,000 European catchments. The catchments included in this study were selected using the following criteria:

- We included only catchments with high-quality delineations, as described by do Nascimento et al. (2024a), to ensure spatial accuracy.
- Only catchments with areas between 50 km² and 35,000 km² were kept, reducing potential aggregation errors in catchment attributes (Beck et al., 2015; Van Dijk et al., 2013; Kratzert et al., 2022).
- We choose catchments with signatures derived from at least 10 years of daily streamflow data (not necessarily consecutive) between 1950 and 2020. This threshold, supported by previous studies (Beck et al., 2015; Kauffeldt et al., 2013), ensured a balance between data availability and temporal representativeness.
- Catchments were excluded if the long-term average streamflow exceeded 10 mm/day or if the runoff ratio was above 1, excluding catchments with potential water balance issues or major anthropogenic impacts.
- Finally, these catchments were nested within a regional river basin with a total area between 7,000 and 35,000 km². This criterion was intended to minimize climatic variability associated with excessively large regional basins, while the second ensured sufficient representation of spatial variability in streamflow signatures.

Following these criterion a total 4,469 catchments remained in the dataset and were used for the different analyses described in Section 2.3.



## 2.2. Geological maps

### 2.2.1. Global geological map

For the global scale geological map, we used the Global Lithological Map (GLiM) (Hartmann et al., 2012). GLiM is a comprehensive database that provides lithological classifications at a scale of 1:3,750,000. The dataset integrates geological information from several regional sources worldwide, including existing geological maps and databases, resulting in a global standardized lithological classification. We used the first level of information, which classifies bedrock-types into 16 major distinct classes (**Table 4**). Despite its lower level of details compared to regional datasets, GLiM's consistency and extensive coverage make it a valuable resource for global-scale studies, and it is used in most existing LSH datasets (Addor et al., 2017;
Helgason and Nijssen, 2024; Höge et al., 2023; Klingler et al., 2021; Kratzert et al., 2022). While its broad classification system may limit detailed geological interpretations, it serves as a foundation for large-scale analyses, providing a balanced trade-off between global coverage and geological detail.

### 2.2.2. Continental geological map

For the continental scale geological map, we used the International Hydrogeological Map of Europe (IHME), version 11, available at www.bgr.bund.de, with a scale of 1:1,500,000 (Duscher et al., 2019; Günther and Duscher, 2019). The IHME map covers almost the entire European continent and parts of the Middle East. We used the third level of detail, which classifies bedrock-types into 31 distinct classes (**Table 4**). The IHME was compiled using the International Geological Map of Europe 1:1,500,000 (IGK1500), maintaining the same scale, topography, and projection.

### 2.2.3. Regional geological map

For the regional scale geological map, we used the same data used by Fenicia and McDonnell (2022). This regional map was specifically developed for the Moselle basin and was therefore used only for the intermediate and small-scale analyses. The Moselle basin spans four countries (54% in France, 37% in Germany, 8% in Luxembourg and 1% in Belgium), with geological maps sourced from different providers depending on the country:

- France: BD LISA database (version 1, niveau 2, ordre 1, scale: 1:250,000, downloaded at https://bdlisa.eaufrance.fr).
- Germany: Geologische Übersichtskarte der Bundesrepublik Deutschland (GÜK200) (scale: 1:200,000, downloaded at www.bgr.bund.de).
- Luxembourg: The map was obtained from the "Administration de la gestion de l'eau" (at a scale of 1:250,000, and available at https://eau.gouvernement.lu/fr.html).
- Belgium: Information from the continental-scale IHME database (see 2.2.2).

Even though the IHME data has a much lower level of details than the other maps, it was kept because less than 1% of the Moselle basin lies in Belgium. The four different maps were combined, resulting in a total of 31 classes over the Moselle (**Table 4**). This combination of data sources provides a more nuanced and region-specific view of geological variability, enhancing the resolution of subsurface properties in the Moselle basin.

## 2.3. Three analyses at progressively smaller scales

We examined three spatial scales, progressively transitioning from larger to smaller regions while increasing the level of analytical detail at each step. This approach aimed to balance depth and breadth, moving from lower-depth, broader-scale analyses to higher-depth, smaller-scale examinations. To ensure clarity, we categorized the spatial scales into three levels:





- Large scale: This level included 63 river basins across Europe, as illustrated in **Figure 1a**.
- Intermediate scale: one selected regional basin, i.e. the Moselle River basin, shown in **Figure 1c**.
• Small scale: Five catchments nested within the Moselle basin (**Figure 1b**).

At each scale, the basins and catchments analyzed included nested sub-catchments, allowing for an examination of their spatial distribution of streamflow signatures. The criteria for selecting catchments at each scale, along with the corresponding methodological approaches for analyzing the spatial distribution of streamflow signatures, are detailed in the following sections.

**2.3.1. Large-scale: 63 basins**

The large-scale analysis made use of the 63 river basins shown in **Figure 1a**. The 63 selected river basins are distributed over a wide spectrum of hydro-climatic and landscape characteristics across Europe. They each contain up to 181 nested sub-catchment streamflow gauges. In total data from 4,469 monitored (sub-)catchments are used in this analysis. See **Table S1** in **supplementary material** for more details. From the 63 river basins of the large-scale analysis, five were selected for more 200 detailed analysis based on their distinct controls on the distribution of streamflow signatures: the Moselle (EStreams ID: DEBU1959), the Cinca (ES000331) and Garonne (FR001604), located on the border between France and Spain, the Vienne (FR003986), located in central west France and the Narew (PL000936), located in northeast Poland. These basins are highlighted in **Figure 1b**.



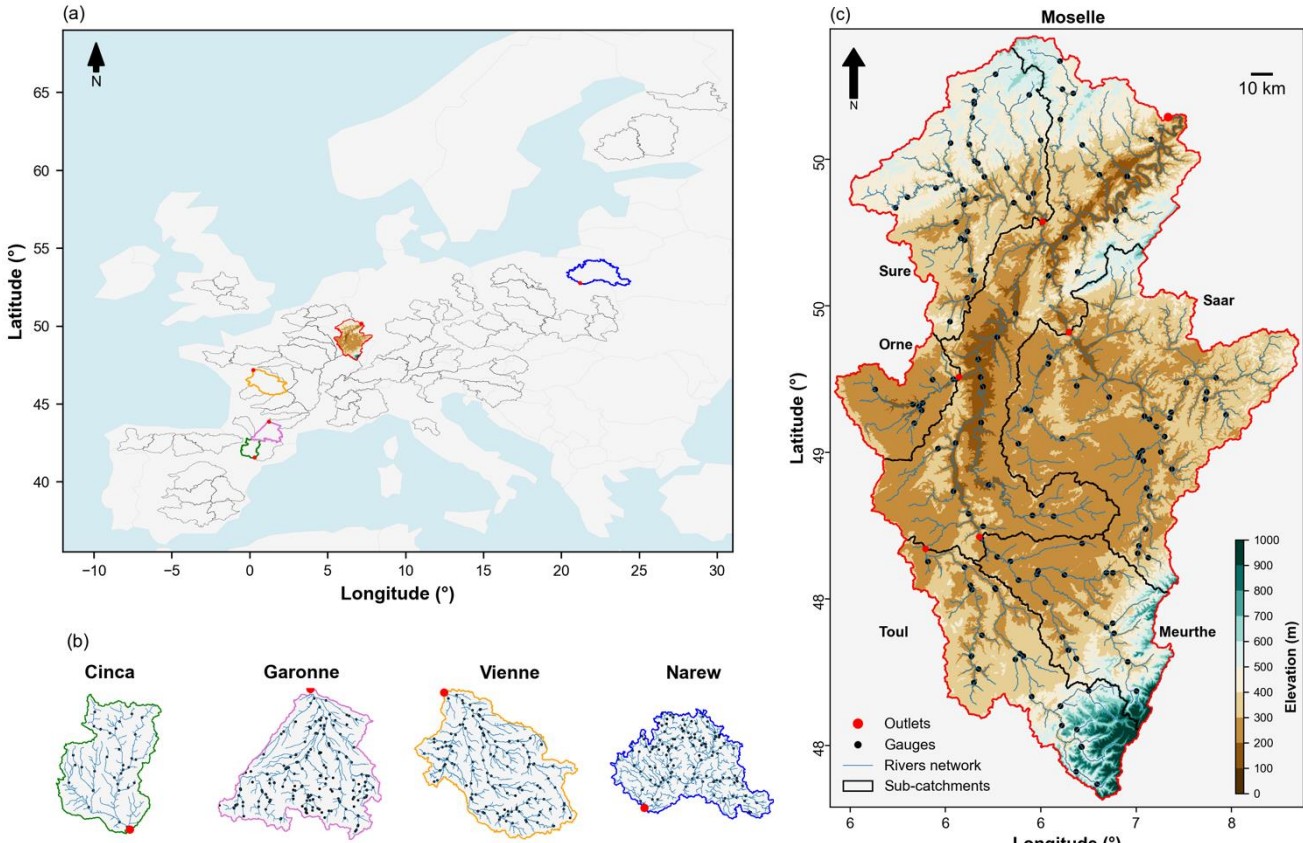

Figure 1: Study area. (a) the 63 river basins used in the large-scale analysis, (b) five selected river basins as examples for the large-scale analysis are highlighted using distinct colors: green (Cinca, Spain, EStreams ID: ES000331), pink (Garonne, France, FR001604), orange (Vienne, France, FR003986), red (Moselle, DEBU1959) and blue (Narew, Poland, PL000936), while the remaining are outlined in black. (c) Moselle river basin also used in the intermediate scale analysis. It also shows the five nested sub-catchments (Moselle-Toul, FR003249; Moselle-Meurthe, FR000159; Moselle-Orne, FR003283; Moselle-Sure, LU000017 and Moselle-Saar, DEBU1957) used in the small-scale analysis. Red circles indicate the outlets of the basins/catchments, while black circles in all panels indicate the locations of the individual sub-catchment outlets at each analysis scale.

### 2.3.2. Intermediate-scale: the Moselle basin

The Moselle basin (**Figure 1c**) was selected for the intermediate scale analysis due to its documented geological influence on streamflow generation (Fenicia and McDonnell, 2022; Hellebrand et al., 2007; Pfister et al., 2017) and the availability of a detailed geological map derived from national databases (Fenicia and McDonnell, 2022). The basin outlet is located at Cochem, approximately 50 km upstream from the confluence with the Rhine in Germany. The 152 gauged sub-catchments used in this analysis are indicated by their outlets in **Figure 1c**.

The Moselle spans 27,100 km², with elevation ranging from 60 to 1,424 m. Its land use is primarily composed of forests (38%), agriculture (30%), and pastures (20%) (Fenicia and McDonnell, 2022). The region experiences annual precipitation between 800 and 1,500 mm/y, while potential evaporation (PET), more consistent across the basin, ranges from 700 to 850 mm/y. Its





substrate exhibits distinct soil heterogeneity, with coarser materials in the south, medium textures in the north, and finer materials concentrated in the central region. The geology of the basin primarily consists of sedimentary and metamorphic rocks, distributed across two geological basins and two massifs (Fenicia and McDonnell, 2022).

### 2.3.3. Small-scale: five nested sub-catchments within the Moselle

For the small-scale analysis, five catchments nested within the Moselle basin were selected (**Figure 1c**). Each catchment was required to contain at least nine gauged sub-catchments to ensure adequate spatial representation. **Table 1** lists their names, outlet codes, area, and the number of gauged sub-catchments. The selected catchments vary in size and number of gauges, with Moselle-Toul being the smallest and Moselle-Saar the largest.

**Table 1: Main overview of the five catchments nested within the Moselle.**

| Name | ID (EStreams) | Area (km$^2$) | Number of sub-catchments |
|---|---|---|---|
| Moselle-Toul | FR003283 | 1,241 | 9 |
| Moselle-Meurthe | FR003249 | 3,396 | 25 |
| Moselle-Orne | FR000159 | 2,883 | 23 |
| Moselle-Sure | LU000017 | 4,255 | 32 |
| Moselle-Saar | DEBU1957 | 6,970 | 32 |

## 3.  Methods

### 3.1.  Descriptors of streamflow, climate and landscape

In this section we present the streamflow, climate, and landscape attributes used in our analyses. To maintain clarity, we use different Greek letters to denote attributes associated with specific domains:

- Streamflow attributes: These are referred to as "signatures" and are identified with the letter σ.
- Climate attributes: These are identified with the letter $\kappa$.
- Landscape attributes: These are classified into five groups based on the maps used to derive them:
  o Topography: $\tau$.
  o Soils: $\xi$.
  o Geology: $\gamma$.
o Land use: $\lambda$.

Each letter is followed by the specific variable name as defined in the EStreams dataset. The attributes used are those readily available from EStreams, which in turn are derived from those commonly found in LSH datasets. An exception is the geology-based attributes, which are described in Section 3.2.



### 3.1.1. Streamflow signatures

We used 6 streamflow signatures to characterize the hydrological behavior of the selected catchments (**Table 2**). These signatures have been selected because they are readily available in LSH datasets and proved useful to characterize different parts of a hydrograph (Addor et al., 2017, 2018; Höge et al., 2023; do Nascimento et al., 2024a). Each signature was originally computed by EStreams using available streamflow data from 1st of October 1950 to the 30th of September of 2022.

**Table 2: Set of streamflow signatures used in this work.**

| Signature | Unit | Description |
|---|---|---|
| $\sigma_{q\_mean}$ | mm day$^{-1}$ | Mean daily streamflow. |
| $\sigma_{slope}$ | - | Slope of the flow duration curve derived using Eq. (3) in Sawicz et al. (2011). |
| $\sigma_{BFI}$ | - | Ratio of mean daily baseflow to mean daily streamflow, hydrograph separation performed using the Ladson et al. (2013) digital filter. |
| $\sigma_{HFD}$ | day of year | Mean half-flow date. It represents the date on which the cumulative streamflow reaches half of the annual discharge. |
| $\sigma_{q\_5}$ | mm day$^{-1}$ | 5 % flow quantile, which represents low flows. |
| $\sigma_{q\_95}$ | mm day$^{-1}$ | 95 % flow quantile, which represents high flows. |

**3.1.2. Climate and landscape attributes**

The set of climate and landscape attributes used in this study is shown in **Table 3**.

**Table 3: Set of catchment attributes used in this work, as described by EStreams** (do Nascimento et al., 2024a)**. The Table divides the attributes into five different groups (i.e. climate: κ, topography: τ, soils: ζ, geology: γ and land use: λ). Geology was split into three sub-groups (i.e., global, continental and regional sources). Similarly to the streamflow signatures, the climatic attributes were**
**derived by EStreams using the available E-OBS time-series dataset between the period of 1950 to 2022.**

| Group | Attribute | Description | Unit | Source |
|---|---|---|---|---|
| Climate | $\kappa_{p\_mean}$ | Mean daily precipitation. | mm day$^{-1}$ | (Cornes et al., 2018) |
| | $\kappa_{pet\_mean}$ | Mean daily potential evapotranspiration (PET). | mm day$^{-1}$ | |
| | $\kappa_{aridity}$ | Ratio between PET and precipitation. | - | |
| | $\kappa_{p\_seasonality}$ | Seasonality and timing of precipitation, which was estimated using the | - | |





| Group | Attribute | Description | Unit | Source |
|---|---|---|---|---|
| | | precipitation and temperature time series, and computed as in (Woods, 2009) | | |
| | $\kappa_{frac\_snow}$ | Fraction of precipitation falling as on days colder than 0 °C. | - | |
| | $\kappa_{hp\_freq}$ | Frequency of P > 5 times the median daily precipitation (high precipitation). | days yr$^{-1}$ | |
| | $\kappa_{hp\_dur}$ | Average duration of periods with consecutive high precipitation events. | days | |
| | $\kappa_{lp\_freq}$ | Frequency of P events < 1 mm day$^{-1}$ (dry days). | days yr$^{-1}$ | |
| | $\kappa_{lp\_dur}$ | Average duration of periods with consecutive dry days. | days | |
| | $\kappa_{sno\_cov\_mean}$ | Mean snow cover percentage over the catchment area derived from satellite. | % | (MODIS/Terra Snow Cover Daily L3 Global 500m SIN Grid, Version 61 [Data Set], 2023) |
| Topography | $\tau_{ele\_mt\_\{max, mean, min\}}$ | Maximum, mean and minimum elevation. | m | (Yamazaki et al., 2019) |
| | $\tau_{slp\_dg\_mean}$ | Mean terrain slope. | ° | |
| | $\tau_{flat\_area\_fra}$ | Percentage of area with slope <3°. | % | |
| | $\tau_{steep\_area\_fra}$ | Percentage of area with slope >15°. | % | |
| | $\tau_{elon\_ratio}$ | Derived elongation ratio (Schumm, 1956) | - | |
| | $\tau_{strm\_dens}$ | Stream density, ratio of lengths of streams and the catchment area. | 1000 Km km$^{-2}$ | |
| Soils | $\zeta_{root\_dep}$ | Depth available for roots. | cm | (ESDD, n.d.; Hiederer, 2013b, a) |
| | $\zeta_{soil\_tawc}$ | Total available water content. | mm | |





| Group | Attribute | Description | Unit | Source |
|---|---|---|---|---|
| | $\zeta_{soil\_fra\_mean\_\{sand, silt, clay, grav\}}$ | Mean sand, silt, clay and gravel fraction of soil material. | % | |
| | $\zeta_{soil\_bd}$ | Bulk density. | g cm$^{-3}$ | |
| | $\zeta_{oc\_fra}$ | Fraction of organic material. | % | |
| | $\zeta_{bedrk\_dep}$ | Depth to bedrock. | m | (Pelletier et al., 2016) |
| Geology {global, continental, regional} | $\gamma_{lit\_\{glob, cont, regi\}\_perm\_low}$ $\gamma_{lit\_\{glob, cont, regi\}\_perm\_med\_low}$ $\gamma_{lit\_\{glob, cont, regi\}\_perm\_med\_high}$ $\gamma_{lit\_\{glob, cont, regi\}\_perm\_high}$ | Fraction of the catchment covered by rock types reclassified under each of the four permeability categories defined (low, medium-low, medium-high and high), as shown in **Table 4**. | - | (Duscher et al., 2019; Günther and Duscher, 2019; Hartmann et al., 2012; AGE, 2024; BDLISA database, 2024; GÜK200, 2024) |
| Land use | $\lambda_{ndvi\_mean}$ | Mean NDVI over the catchment area. | - | (MODIS/Terra Vegetation Indices 16-Day L3 Global 500m SIN Grid V061 [Data set], 2023) |
| | $\lambda_{lai\_mean}$ | Mean LAI over the catchment area. | - | (MODIS/Terra Leaf Area Index/FPAR 8-Day L4 Global 500m SIN Grid V061 [Data set], 2023) |
| | $\lambda_{lulc\_2006\_urban}$ | Sum of the aggregated fractions of classes 111 to 124. | - | (CORINE Land Cover — Copernicus Land |
| | $\lambda_{lulc\_2006\_agriculture}$ | Sum of the aggregated fractions of classes 212 to 300. | | |



| Group | Attribute | Description | Unit | Source |
|---|---|---|---|---|
| | $\lambda_{lulc\_2006\_forest}$ | Sum of the aggregated fractions of classes 300 to 314. | | Monitoring Service, 2023) |
| | $\lambda_{lulc\_2006\_grass}$ | Sum of the aggregated fractions of classes 315 to 400. | | |

### 3.2. Reclassification of the geological maps

Lithological maps, in their raw form, are challenging to use directly for regression analysis with streamflow signatures. To enable their use, lithological classes must be assigned numerical values that reflect their hydraulic properties. To address this, we reclassified the geology into four relative permeability classes: low, medium-low, medium-high, and high, following the

approach introduced by Fenicia and McDonnell (2022) and supported by additional references from the literature (Bagdassarov, 2021; Gleeson et al., 2011).

**Table 4** shows the details of this reclassification, associating each geological map class with its corresponding permeability category. For instance, volcanic and plutonic rocks were assigned to the "low permeability" class, while unconsolidated sediments, such as conglomerates, were categorized as having "high permeability". **Figure 2** shows the spatial distribution of

these categories from the global, continental and regional geology maps across the Moselle basin. The three geological maps generally agree on permeability distribution, though discrepancies appear in the northern area.

**Table 4: Reclassification of the global, continental and regional geology classes into four permeability classes. The table also shows the sources of each map.**

| Permeability | Global | Continental | Regional |
|---|---|---|---|
| Low | Evaporites, Ice & glaciers, Acid plutonic rocks, Basic plutonic rocks, Acid volcanic rocks and Basic volcanic rocks | Plutonic rocks, Volcanic rocks, Inland water, Snow field / ice field, Clays, Quartzites, Shales | Crystallin basement, Plutonic rock, Quartzite, Schist, Volcanic rock |
| Medium-low | Metamorphic, Intermediate plutonic rocks, Pyroclastic and Intermediate volcanic rocks | Claystone & clays, Marbles, Marls, Marlstones, Marlstones & clays, Marlstones & marls, Phyllites, Schists, Gneisses, Silts | Arkoses, Dolomite rocks, Limestone & marls, Marls, Marls & dolomites, Marls & limestones, Marl & sandstones, Sandstone & siltstone, Sandstone, siltstone & schists, Schist & sandstones, Silt, Silt & schist, Siltstone, sandstone and Schist |



| Medium-high | Carbonate sedimentary rocks and Mixed sedimentary rocks | Conglomerates & clays, Limestones, Limestones & sands, Sandstones & clays, Sandstones & marls, Limestones & clays, Limestones & marls, Marlstones & sands | Limestones, Sandstones & marls, Sandstones & schists, Sandstones, conglomerates & marls |
|---|---|---|---|
| High | Siliciclastic sedimentary rocks and Unconsolidated sediments | Conglomerates, Conglomerates & sands, Gravels, Sands, Sandstones, Sandstones & Sands | Alluvium, Coal, Conglomerates, Gravel & sand, Sand, Sand & Gravel, Sandstone & conglomerates and Sandstones |
| Source | GLiM database, level 1 (Hartmann et al., 2012) | IHME 1500, level 3 (Duscher et al., 2019; Günther & Duscher, 2019) | (AGE, 2024; BDLISA database, 2024; Duscher et al., 2019; GÜK200, 2024; Günther & Duscher, 2019) |



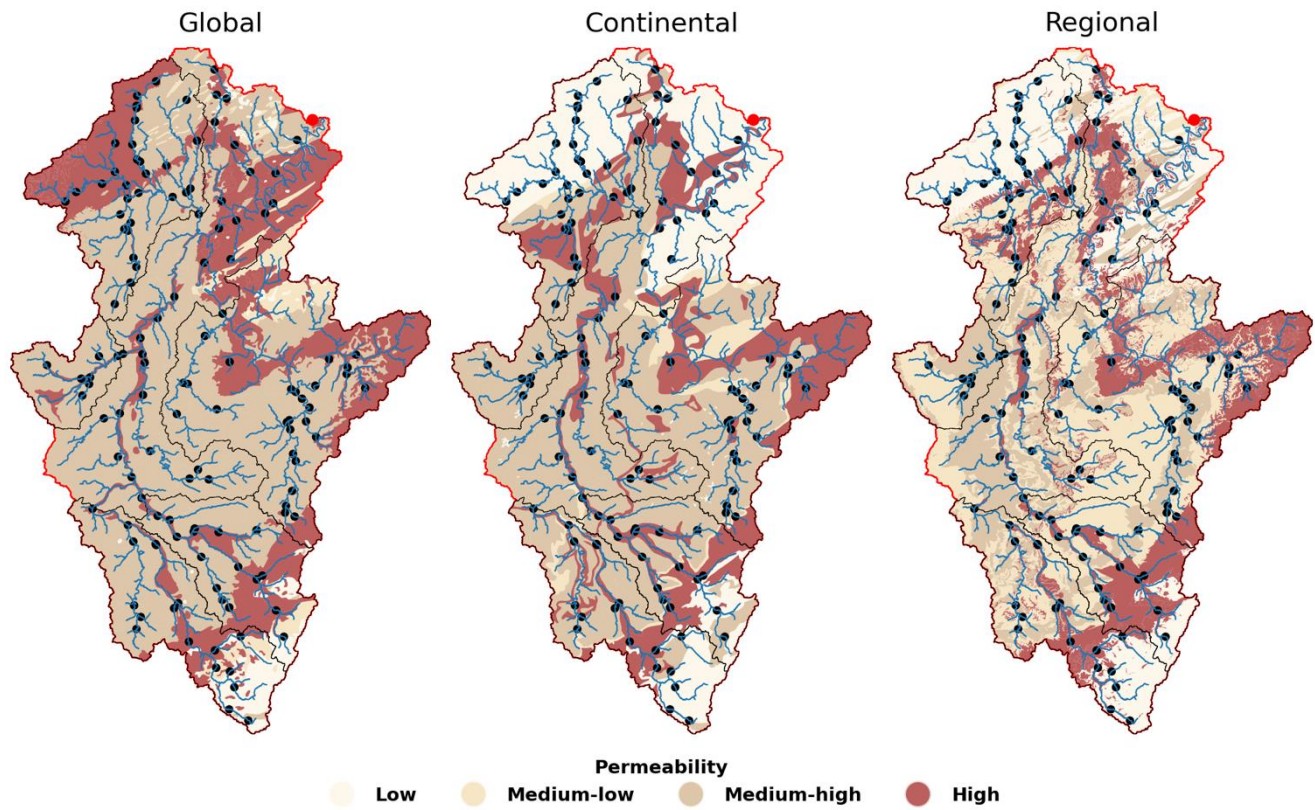

**Figure 2: Spatial distribution of the four permeability categories across the Moselle basin for the global, continental and regional geology maps. In background there is the river network in blue, the streamflow gauges (sub-catchments) as black dots, and the basin outlet in red.**

The four permeability classes were represented by attributes defining the spatial coverage of each permeability class relative to the total catchment area. For instance, the variable $\gamma_{lit\_cont\_perm\_high}$ in **Table 3** Indicates the relative area fraction of high-permeability zones in the continental map, therefore combining the areas of associated rock-types, e.g., conglomerates, conglomerates & sands, gravels, sands, sandstones, and sandstones & sands (**Table 4**).

### 3.3.  General procedure for identifying dominant controls on streamflow

In this study, we conducted an exploratory statistical analysis to identify dominant climate or landscape attributes influencing streamflow generation across different scales. This was done based on the Spearman correlation coefficient ($r_s$) between each catchment attribute (listed in **Table 3**) and each streamflow signature (**Table 2**). To analyze the impact of different geological maps, the analysis was performed separately using data of the corresponding attributed from each map—global, continental, and regional—. This approach allows us to evaluate the influence of geological data level of detail on correlation outcomes and assess whether observed correlations might indicate causal relationships. While we acknowledge that more sophisticated machine learning techniques (Addor et al., 2018; Beck et al., 2015; Kuentz et al., 2017), could enhance predictive power for streamflow signatures, these methods are much more complex than our approach, and do not necessarily result in improved





hydrological understanding. We also note that many climate and landscape attributes are interdependent (Mathai and Mujumdar, 2019), whereas our approach allows to assess only individual correlations. However, the primary objective of this study is to identify the dominant controls on streamflow signatures, ensure physical interpretability, and specifically compare geological maps of varying levels of detail, rather than focusing on maximizing predictive power.

### 3.4. Specific procedure

#### 3.4.1. Large-scale

For the 63 regional basins, we followed this procedure:

- First, we calculated the $r_s$ between each of the 6 streamflow signatures (**Table 2**) and the 47 catchment attributes (**Table 3**) using all available sub-catchments within each basin. For example, for the Moselle basin, we used data from the 152 sub-catchments to compute each $r_s$ value. Overall, this step resulted in a total of 6 x 47 x 63 = 17,766 correlation values.
- To provide a broad overview of landscape controls on each signature by reducing dimensionality and streamlining interpretation, we only reported the maximum $|r_s|$ for each of the six attribute groups listed in **Table 3**: climate, topography, soils, land use, global geology, and continental geology. This step narrowed the correlation values to 6 x 6 x 63 = 2,263.

#### 3.4.2. Intermediate scale

For the Moselle basin, we applied the same procedure as described above for the large-scale analysis, with the addition of data from the regional geology map.

#### 3.4.3. Small-scale

For the five catchments of the Moselle, we employed the following methodology:

- We calculated the correlation between the 6 streamflow signatures and 47 catchment attributes for each of the five catchments of the Moselle, using all available nested sub-catchments within each catchment. This resulted in a total of 6 x 47 x 5 =1.410 correlation values $r_s$.
- Instead of selecting only the maximum $|r_s|$ value per group, we conducted a more refined analysis here, focusing on the physical coherence of correlations, consistent with the hypotheses raised in Section 3.1.

This more in-depth analysis aimed to assess how local factors might alter the relationships identified at broader scales, allowing us to distinguish correlations that are likely to reflect true causal relationships.

## 4. Results

### 4.1. Large scale analysis

#### 4.1.1. Correlation dynamics for baseflow index

Here, we focus the analysis on the representation of the baseflow index ($\sigma_{BFI}$), a streamflow signature commonly associated with groundwater flow, where geology is expected to play a major role. **Figure 3** shows the maximum absolute correlation



values for $\sigma_{BFI}$ across the 63 basins, with each attribute group represented by a distinct color. The basins are sorted based on ascending maximum correlation between $\sigma_{BFI}$ and global geology attributes.

The results shown in **Figure 3** indicate that most basins have at least one attribute that provides relatively high correlation with baseflow index. We observe that 75% of the basins (47 out of 63) exhibited at least one attribute group with $|r_s| > 0.50$. At the same time, basins behave very differently in terms of which property best explains the variability of baseflow, as well as in the relative ranking of these attributes. In most cases, land use (15) was the dominant control, followed by geology (13), soils (13), climate (12), and topography (10). Particularly, landscape attributes show stronger correlations with $\sigma_{BFI}$ than climatic

attributes in 51 out of the 63 catchments. This result highlights the variability in dominant controls on $\sigma_{BFI}$ in the 63 studied basins.

Comparing the maps of different detail, the continental map with average $|r_s| = 0.42$ appears slightly more informative than the global map with average $|r_s| = 0.40$. **Figure 3** highlights that the most striking increases are in the left side of the plot, representing the basins with $|r_s| < 0.50$ for the global maps. In cases where $|r_s| > 0.50$ the increases are more ambiguous. In

such cases, generally there is not much difference between the two geological maps, highlighting the added value of the continental map for basins performing very poorly with the global map. This highlights the complex and variable role of landscape attributes in controlling baseflow signatures across basins. The continental map leads on average to higher correlations than the global map, although the global map appears to be superior in some cases, which warrants a closer examination.

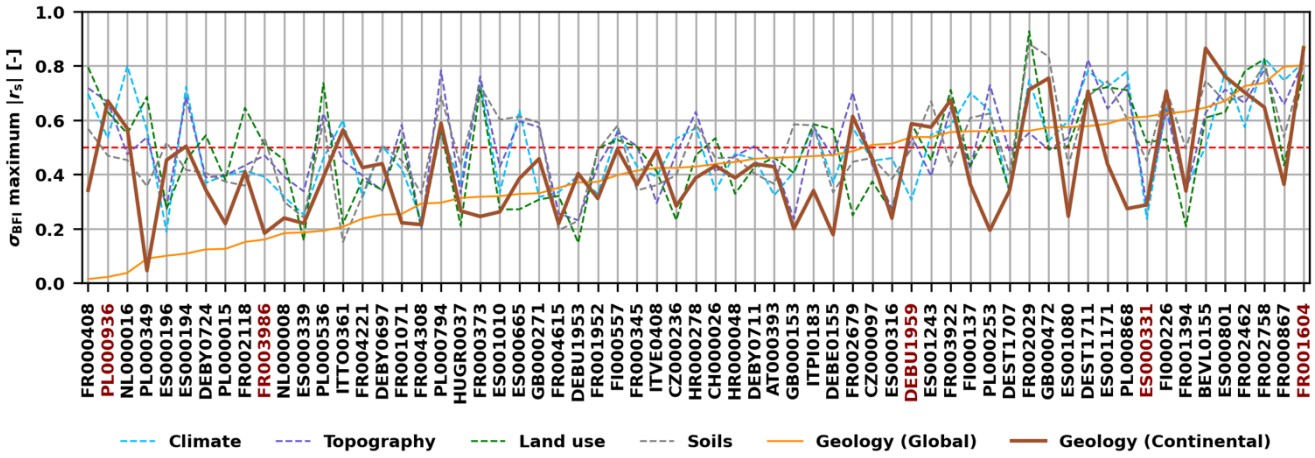


Figure 3: Maximums $|r_s|$ for each catchment attribute group for each river basin. Each color represents the respective maximums $|r_s|$ for a specific catchment attribute group (e.g., climate is shown in blue). The IDs of the Cinca (ES000331), Garonne (FR001604), Vienne (FR003986), Moselle (DEBU1959) and Narew (PL000936) basins are indicated in red. To facilitate interpretation, the red dashed line represents the $|r_s|$ of 0.50.

**4.1.2. Signature correlations between geology maps**

**Figure 4** shows a more detailed comparison between the global and continental map across a larger set of signatures. It reports the maximum $|r_s|$ values of all geology attributes for each of the used streamflow signatures derived from the global and continental geology maps. Each blue dot represents one basin, with the five selected basins highlighted in different colors.





First, it can be seen the $|r_s|$ values between geological attributes and hydrological signatures generally increase when transitioning from global to continental derived attributes, as indicated by most points being above the diagonal line: $\sigma_{q\_mean}$ (34), $\sigma_{slope}$ (33), $\sigma_{BFI}$ (32), $\sigma_{HFD}$ (37), $\sigma_{q\_5}$ (32), $\sigma_{q\_95}$ (41). The attributes showing the highest correlation varied (often high or medium permeability). The signatures $\sigma_{q\_mean}$ (**Figure 4a**) and $\sigma_{q\_95}$ (**Figure 4f**) exhibited most of their basins concentrated around or above the diagonal line, indicating the greatest increases among the signatures. The Moselle basin, in particular, showed improvements with, on average, $|r_s| = 0.45$ using the global map increasing to $|r_s| = 0.60$ using the continental maps across all six signatures.

Moreover, most of the cases where the global map showed a higher correlation than the continental were cases where $|r_s|$ for both geological maps was already low, as can be seen by such cases being mostly found in the $|r_s| < 0.50$ part of the graphs.

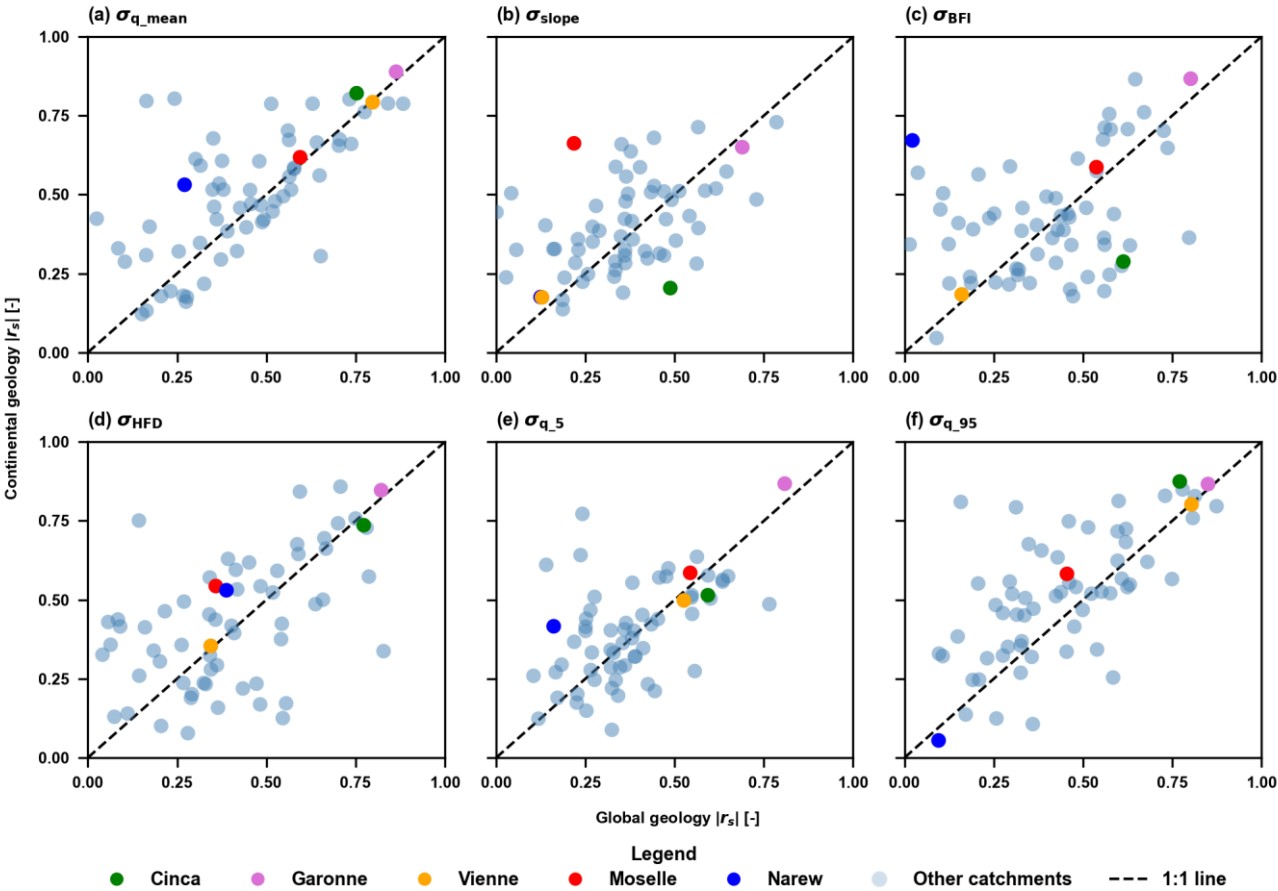

**Figure 4: Scatter plot of the $|r_s|$ derived from the global geology map, versus the $|r_s|$ from the continental geology map for the six streamflow signatures. Each light blue circle represents one of the 63 river basins. The 1:1 line is shown in dashed black.**

Focusing only on basins with $|r_s|$ differences ±0.1 between global and continental map attributes (about 30 for each signature), the findings revealed consistent increases in more than 50% of these basins. For instance, $\sigma_{q\_mean}$ showed increases in 85% of



the basins, while $\sigma_{q\_95}$ in 81%. $\sigma_{slope}$ and $\sigma_{q\_5}$ increased in 69% of the basins, while $\sigma_{HFD}$ in 63% and $\sigma_{BFI}$ in 59%. These results underscore the added value of continental-scale geology maps in capturing spatial hydrological characteristics, particularly for flow extremes and surface water dynamics.

### 4.1.3. Further exploration of five representative basins

To gain deeper insights into disparities in results arising from distinct underlying geological maps, five selected basins with distinct correlation patterns (**Figure 2** and **Figure 5**) were analyzed, namely the Moselle (DEBU1959), Cinca (ES000331), Garonne (FR001604), Vienne (FR003986) and Narew (PL000936). These basins exhibit a range of behaviors, from high and consistent correlations in the Garonne to substantial differences between global and continental maps in the Cinca, Narew, and Moselle, and low correlations in the Vienne irrespective of the map used. The key variations in these correlations are linked to differences in geological classification, spatial heterogeneity, and the level of fine boundary contours (spatial resolution) in each map. A detailed description of their raw geological classifications is provided in **Appendix A**. From this analysis it was found that:

For the Moselle basin (**Figure 2**), our results show that the continental map provided with $|r_s|$ = 0.59 a slightly better representation of the baseflow compared to the global map with $|r_s|$ = 0.54 (**Figure 4c**). **Figure 2** highlights that the main difference between the two maps is in the northern area of the basin, where the global map classified a substantial portion as siliciclastic rocks (high permeability in our classification). The continental map, however, provided a more detailed differentiation, identifying shale (low permeability), which better aligned with the observed low $\sigma_{BFI}$ values in the area. This means that using the global maps, our reclassification assigned 3.8% of the basin to low permeability, while the continental map reclassified 22.7% to low permeability.

The Cinca basin shows an interesting pattern. While the global geology map presented a $|r_s|$ = 0.60 between baseflow and geology, the continental map yielded a much lower correlation with $|r_s|$ = 0.29 (**Figure 4c**). **Figure 5** shows that the global map provided a higher spatial variability alongside with finer details than the continental map. In fact, the continental map expressed a sharp change in the permeabilities from the more upstream area to the downstream area.

For the Garonne basin, both geology maps displayed very high correlation values ($|r_s|$ > 0.80, **Figure 4c**). **Figure 5** indicates that both maps classified this basin with dominant rock types such as mixed sedimentary rocks and unconsolidated sediments, which yield also high permeability (**Appendix A**). Additionally, both maps present a strong spatial geology variability (gradient) over the whole basin area (**Figure 5**). In quantitative terms, the global map classified 29% and 52% of the area respectively as high and medium-high permeability, and the continental with 17% (high), 49% (medium-high), demonstrating a good balance among the used reclassified classes.

The Vienne basin is characterized by with very low correlations ($|r_s|$ < 0.20) for both geology maps (**Figure 4c**). **Figure 5** reveals that both maps agreed about a clear separation of low-permeability rock types in the upstream area, and high-permeability in the downstream. There is thus not a smooth geological variability gradient that could explain spatial differences in baseflow, but a sudden change from upstream to downstream in the pattern of the reclassified permeabilities. This is a similar pattern to the continental map of the Cinca basin. Notably, **Figure 3** shows that the highest $|r_s|$ using the climate, topography, land-use and soil attributes remained similarly low in the Vienne with $|r_s|$ < 0.55.

For the Narew basin, the continental map showed a pronounced $|r_s|$ =0.70, while the global map suggested near-zero correlation. **Figure 5** shows that the permeability reclassification provided by the global map classified most of the basin (95% of its area) with high permeability, whereas the continental map distinguished the basin into areas of high (45%) and medium-low (55%). As can be seen in **Appendix A**, the main difference in the reclassifications came from the limited level of detail in the global map, which classified most of the area simply as unconsolidated sediments.




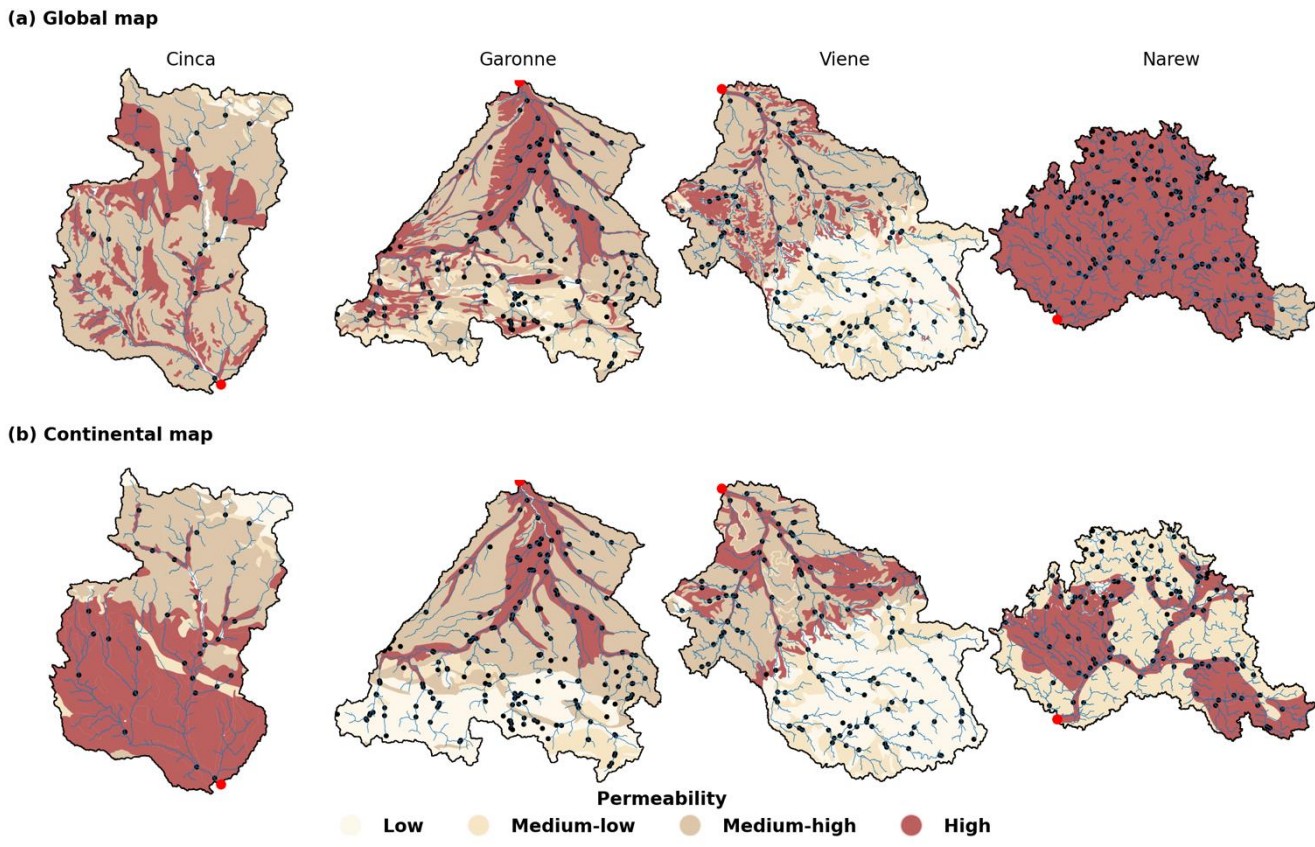

**Figure 5: Spatial distribution of the four permeability categories across four selected basins for the (a) global and (b) continental geology maps. The Moselle was excluded here because it is already depicted in Figure 2. In background there is the river network in blue, the streamflow gauges (sub-catchment outlets) in black, and the respective basins outlets in red.**

Overall, the global and continental maps appear to show a trade-off in their ability to accurately reflect geological classes. While the continental map offers a broader range of geological classes, allowing for more precise classification of hydrological properties, it tends to be more approximate in representing the boundaries between different layers. These boundaries are more jagged in the global map and smoother in the continental map, indicating a compromise between class diversification and the level of fine boundary contours. With exception of the Narew, this was evident for all other four basins explored here. Although such higher level of fine contour detail not always provided higher correlation (Moselle, Garonne and Vienne), it likely explains the higher correlation of the global map for the Cinca basin (**Figure 5**).

## 4.2. Intermediate scale analysis

Here we present the results of our analysis on the Moselle basin, incorporating the regional geology map besides the global and continental maps. The heatmap in **Figure 6** shows the maximum $|r_s|$ values for the six streamflow signatures across the seven attribute groups within the basin.



First, **Figure 6** suggests a consistent increase in $|r_s|$ values from global to continental to regional geology maps. In some cases, correlations increased considerably: $\sigma_{slope}$ increased from 0.22 to 0.66 to 0.70. In other cases, the increase was more modest: $\sigma_{q\_mean}$ increased from 0.59 to 0.62 to 0.66. Generally, the impact of geology appears more pronounced for signatures influencing the shape of the hydrograph rather than the magnitude of streamflow.

Additionally, **Figure 6** illustrated that when using the global geology map, geology appears to have less influence than other landscape or climate controls across all signatures. The continental map shows geology as a dominant control only for $\sigma_{slope}$, whereas its influence remains equal to or lower than that of landscape or climate characteristics. The regional map highlights geology as the primary control for four out of six streamflow signatures, excluding $\sigma_{q\_mean}$ and $\sigma_{q\_95}$.

This progression—from very limited (global) over moderate (continental, affecting one signature), to strong influence (regional, affecting four signatures)—demonstrates the increasing relevance of geology in streamflow behavior as the quality of geology information improves. The affected signatures are consistent with expectations: those related to baseflow, and flow persistency show stronger geological influence, whereas $\sigma_{q\_mean}$ (linked to flow magnitude) is more dependent on climate, and $\sigma_{q\_95}$ (associated with peak flows) is primarily influenced by topography.

**Appendix B** further illustrates the higher correlations of the regional geology map for $\sigma_{slope}$, $\sigma_{BFI}$ and $\sigma_{HFD}$ . The scatter plots show that the global map performs poorly mainly in smaller catchments, where geology is either oversimplified into a single category or lacks representation for certain classes. This suggests that its lower correlations stem from its less detailed classification.

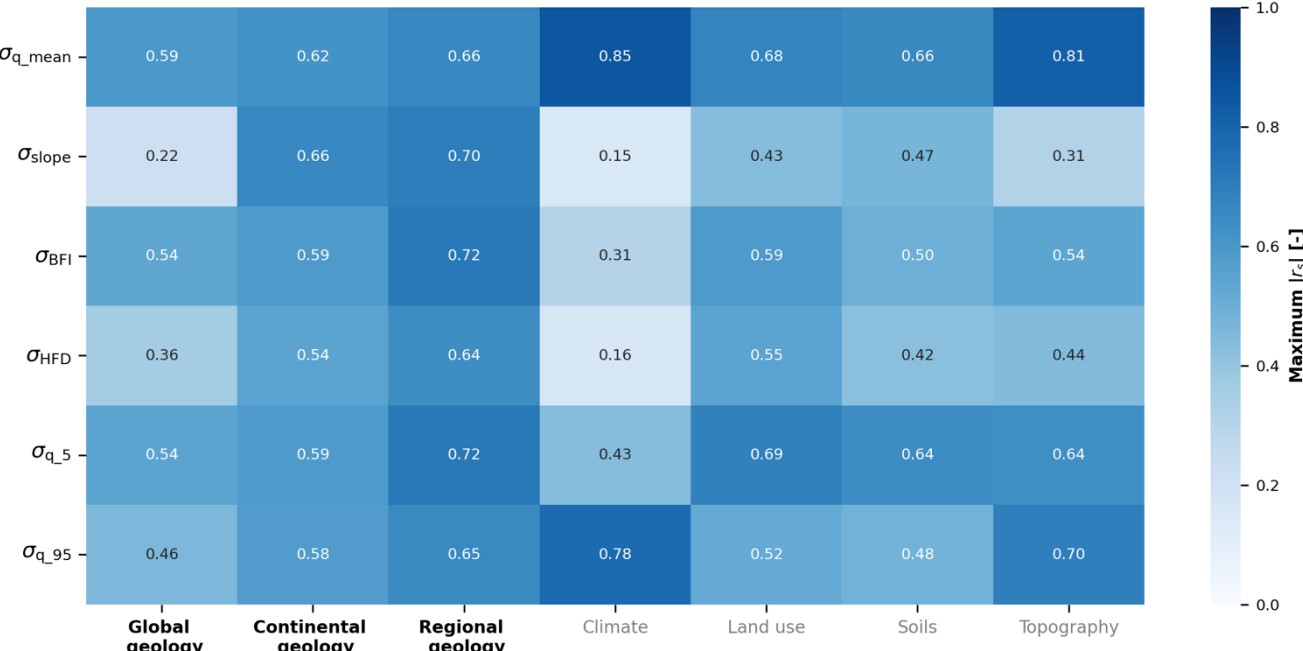

**Figure 6: Heatmap of maximum $|r_s|$ for the set of streamflow signatures in relation to the seven attribute groups used in this work. Notice that the $|r_s|$ values are in absolute terms for improving visualization, ranging from zero to one.**





### 4.3. Small scale analysis in the Moselle sub-catchments

**4.3.1. Spatial patterns of hydrological variability**

This section provides an overview of hydrological variability across the Moselle basin and its five selected catchments, serving as a foundation for interpretation. **Figure 7** illustrates patterns of key climatic, landscape and streamflow characteristics. The figure highlights substantial variability in all considered properties across the Moselle. Even neighboring catchments can exhibit markedly different responses. For example, Moselle-Toul and Moselle-Meurthe, despite being geographically adjacent 440 and similar in size, display different hydrological patterns.

The mean flow is the highest in the Southern part of the Moselle (**Figure 7a**), reaching $\sigma_{q\_mean} > 2$ mm/day across most of the Moselle-Toul. In contrast, the lowest $\sigma_{q\_mean}$ along with reduced spatial variability are found in the central part of the Moselle: Orne and Saar sub-catchments.

The spatial pattern of $\sigma_{q\_mean}$ closely resembles that of mean precipitation ($\kappa_{p\_mean}$, **Figure 7b**), suggesting a close relation 445 between the two variables—specifically, that, unsurprisingly, mean discharge is primarily controlled by mean precipitation (**Figure 7a**). In turn, precipitation variability appears to relate to elevation variability, as indicated by the similarities between precipitation patterns and elevation (**Figure 1b**) or mean terrain slope ($\tau_{slp\_dg\_mean}$; **Figure 7c**).

The baseflow ($\sigma_{BFI}$), shows a very different pattern than $\sigma_{q\_mean}$ (**Figure 7d**). Its highest values and variability are found in Moselle-Saar, while Toul and Sure presented the lowest $\sigma_{BFI}$ values and reduced variability. These patterns align with the 450 geological classification shown in the Moselle geology map (**Figure 2**), where higher permeability in the southeast corresponds to increased baseflow, while lower permeability in the north results in reduced baseflow.

Rooting depth (**Figure 7e**) is shallowest in the Northern part of the Moselle (Moselle-Sure), an area that also has lower forest cover (**Figure 7f**) compared to the rest of the basin. According to Fenicia and McDonnell (2022), this area is also strongly influenced by agriculture. Notably, from the visual inspection, it seems that both attributes seem to mirror the baseflow pattern 455 in the Sure catchment.



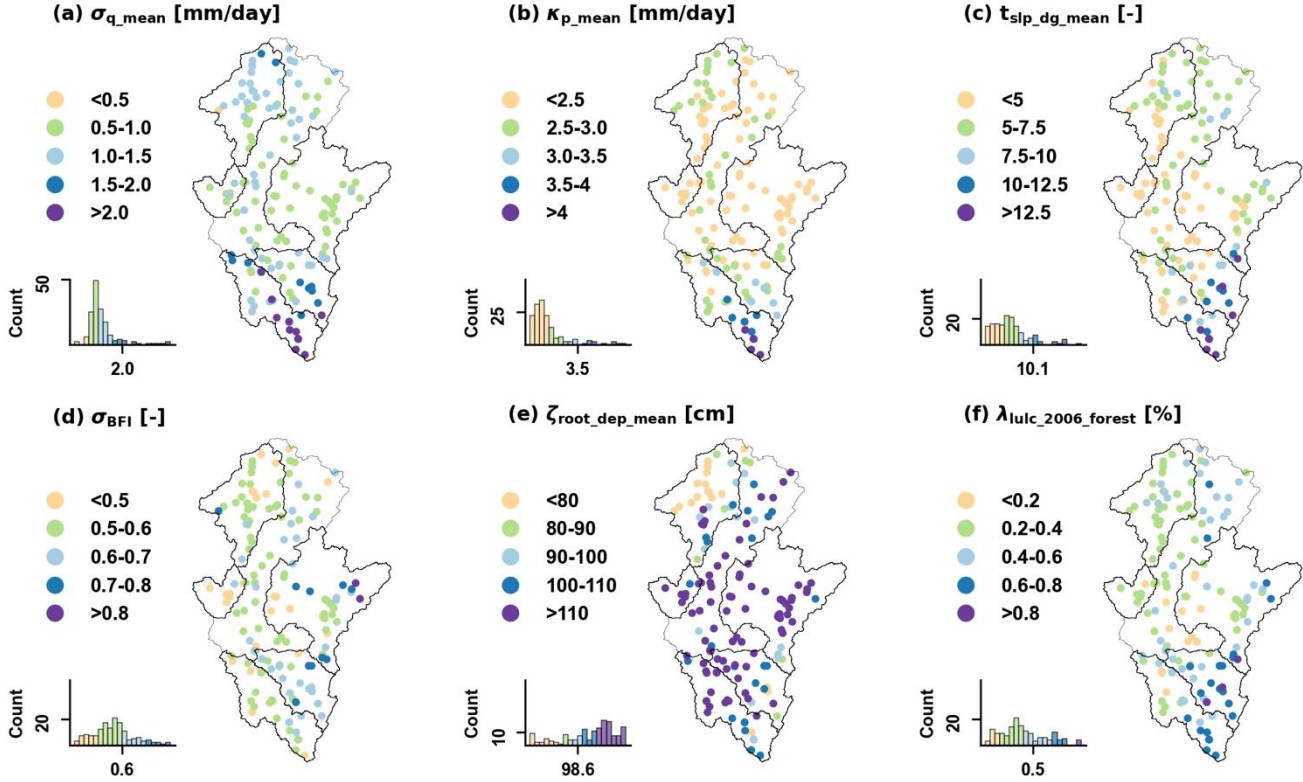

**Figure 7: Spatial variability of selected streamflow signatures and landscape attributes used in this work over the Moselle basin. Each subplot shows the values of each variable with colors corresponding to specific intervals. Note that each subplot also shows the histogram of the specific variable, with histogram bins colored to match the ranges used in the map circles.**

### 4.3.2. Baseflow index variability across the five Moselle catchments

We conducted a detailed analysis of baseflow variability across the five sub-catchments to investigate its relationship with landscape and climatic attributes, and to identify potential differences in hydrological behavior. **Figure 8** presents a heatmap displaying the $r_s$ values between $\sigma_{BFI}$ and key climate and landscape attributes. Unlike **Figure 6**, which grouped attributes into broader categories, **Figure 8** displays individual attributes, focusing on a subset of the original 47 while retaining the correlation signs.

Correlation patterns vary significantly across catchments, as shown by the diverse pattern of the columns of **Figure 8**. The same attribute can exhibit strong correlation in one catchment and weak in another or even change correlation sign. Toul, Meurthe, and Saar consistently show stable correlation signs, though with varying magnitudes. Orne and Sure, in contrast, exhibit distinct correlation patterns both from each other and from the other catchments.



The dominant correlating variables vary by catchment. In the Orne it is mean NDVI ($r_s$ = -0.88), in the Toul it is the medium-low global permeability ($r_s$ = 0.90), in the Meurthe it is agriculture cover ($r_s$ = -0.94), in the Sure it is the fraction of flat area ($r_s$ = 0.71), in the Saar it is the medium-high continental permeability ($r_s$ = -0.91).

Geological maps show distinct correlation behaviors. For example, in the Orne catchment, the global geology map shows no correlation with $\sigma_{BFI}$, whereas the regional map reached a much stronger one to $\gamma_{lit\_cont\_perm\_med\_low}$ ($r_s$ = -0.77). In the Toul
catchment, the global map shows a very high correlation to $\gamma_{lit\_glob\_perm\_med\_low}$ ($r_s$ = 0.90), while the regional map indicates a weaker correlation to $\gamma_{lit\_cont\_perm\_med\_high}$ ($r_s$ = -0.65).

Overall, the regional geological map provides the most stable correlations. Specifically, high regional permeability always shows positive correlations ($r_s$ > 0.40). A similar pattern can be observed for medium-low regional permeability ($r_s$ < -0.47), which only presented a positive sign for Sure. The global geological map, by contrast, displays more variation in both
correlation sign and magnitude. While it contains many near-zero correlations (Orne, Meurthe, and Sure), it also presents extreme values (Toul).

The regional geological map aligns best with hypothesis 2, which states that baseflow should show positive correlations with high-permeability geological attributes (section 1). The global map's very high positive correlation ($r_s$ = 0.93 for $\gamma_{lit\_glob\_perm\_med\_low}$) contradicts this hypothesis.

Notably, strong correlations do not necessarily imply causality. Hence these results should serve as a basis for interpretation alongside process-based understanding.





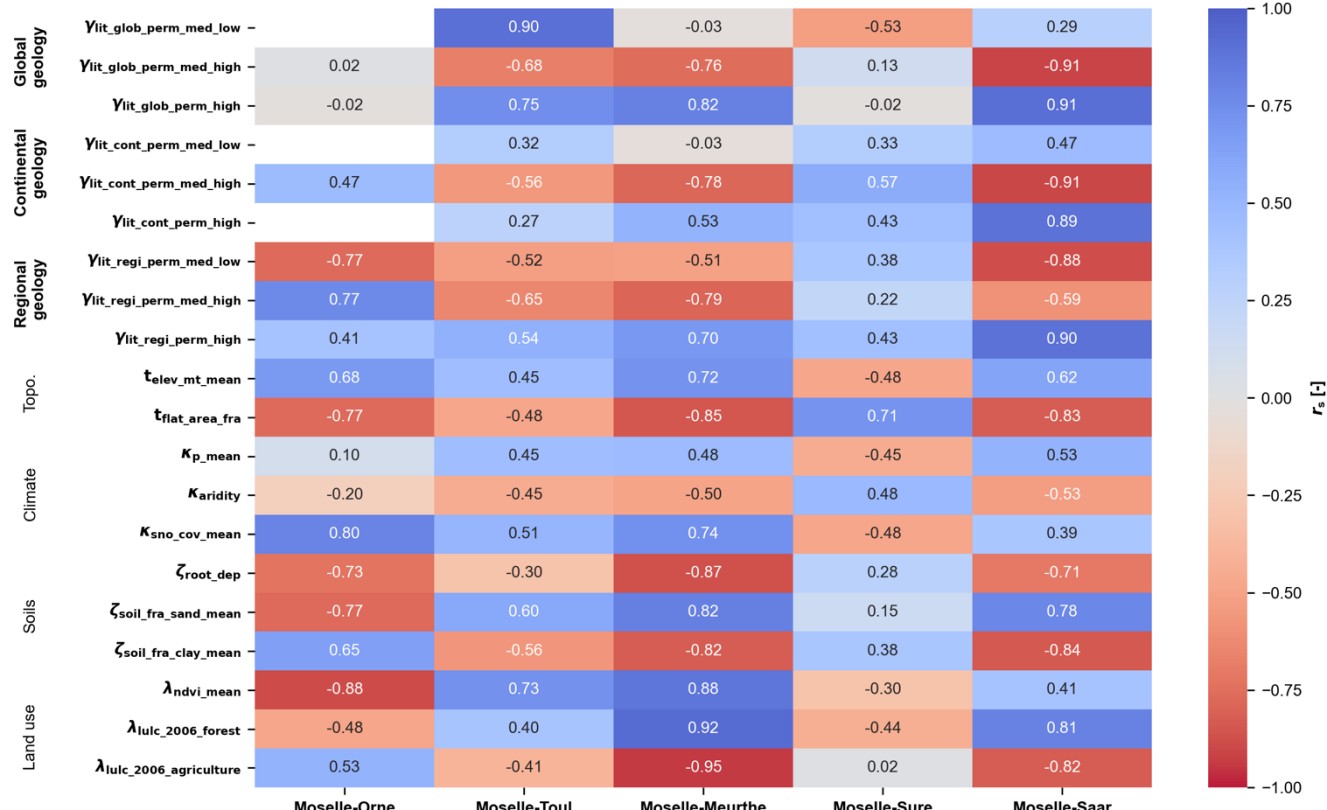

**Figure 8: Heatmap representing the $r_s$ between the $\sigma_{BFI}$, and a set of the different catchment attributes used in this work for each of the five sub-catchments of the Moselle. The attributes are divided into geology (sub-divided into global, continental and regional), topography, climate, soils and land use.**

## 5. Discussion

### 5.1. Assessing the role of geology in streamflow signatures across scales

The large-scale analysis reveals considerable variability in the relationships between landscape attributes and streamflow signatures. Unlike previous studies, which often found that climate exerted a stronger influence than landscape characteristics studies (Addor et al., 2018; Beck et al., 2015; Kratzert et al., 2019; Kuentz et al., 2017), our results indicate the opposite—landscape attributes generally exhibit stronger correlations than climate variables. This discrepancy may be due to the scale of analysis, which focused on individual catchments rather than aggregating them, potentially allowing local geological and topographical controls to emerge more distinctly.

Regarding the two geological maps, the continental one generally yields higher correlations than the global map. Notably, for groundwater-oriented streamflow signatures—which may be expected to be strongly influenced by geology—the continental or global maps do not always produce the highest correlations, which sometimes is reached by climate or other landscape





attributes. Whether this discrepancy arises from fundamentally different hydrological controls or reflects limitations in the information content of the global geology map remains an open question, as further discussed in Sections 5.3 and 5.4.

In our view, the global and continental geology maps have complementary strengths and weaknesses. The global map appears to capture small-scale geological features more effectively, as seen in its more intricate and jagged class boundaries (e.g., in **Figure 5a** compared to **Figure 5b**). This allows for a more precise delineation of geological units. However, the continental map offers a greater number of geological classes, enabling a finer and more accurate classification of relative permeabilities. This enhanced classification allows for better differentiation of geological influences on hydrological processes (see Section 5.3.2).

The ability to represent basin heterogeneity, mostly correlated to the number of geological classes provided in the first place, seems to matter more when it comes to the strength of the correlations found. However, there are exceptions, as discussed in Section 5.2. Ideally, the regional map used in the intermediate-scale analysis should resolve the shortcomings of both global and continental maps by providing even more detailed geological information.

At the intermediate scale, the analysis within the Moselle basin shows a clear gradient of improvement in predicting streamflow signatures when transitioning from global to continental to regional geology maps. This pattern reinforces the hypothesis that more detailed geological maps provide more useful information for hydrological analysis. The variability in results is significant, even to the extent that different geology maps lead to contrasting interpretations of hydrological understanding. When using the global map, geology appears to have little to no control over streamflow signatures, with some signatures, particularly related to baseflow, being poorly predicted. However, the use of continental and regional maps overturns these 520 initial perceptions, demonstrating that geology indeed plays a crucial role, particularly in controlling streamflow signatures related to baseflow generation.

At the smallest scale considered, different hydrological controls emerge as significant for different sub-catchments when predicting streamflow signatures. Similar to the global-scale analysis, there is considerable variability in determining which attributes best correlate with specific signatures across catchments. However, the regional geology map stands out by 525 consistently providing meaningful and interpretable results. The correlations obtained using the regional map are not only relatively high but also align with expected hydrological behaviors (hypothesis 2 in section 1). This suggests that the regional geology map leads to results that are both reliable and interpretable, leading to physically meaningful insights.

Overall, these findings highlight the challenges in catchment hydrology and large-sample studies in transitioning from correlation to causality. While some attributes exhibit strong correlations with certain streamflow signatures, these 530 relationships may sometimes be coincidental rather than truly causal. Establishing causality requires detailed process-based analyses, a level of scrutiny that is often incompatible with the scale of large-sample studies. These results emphasize the importance of complementing large-scale statistical approaches with detailed hydrological insights to improve our understanding of hydrological controls.

### 5.2. Need for enough geological heterogeneity and fine detail

To assess the influence of a landscape or climate attribute on a streamflow signature, the attribute must first exhibit variability. In the case of geology, its effect on streamflow signatures can only be evaluated if geological attributes vary across the study area. This condition is not always met in all case studies considered. Our results indicate that regardless of the geological level of detail, high correlations require sufficient geological heterogeneity (i.e., spatial geological gradient). In regions with homogeneous geology, even detailed maps provide limited added value. This is evident in cases where geology exhibited weak 540 correlations with streamflow signatures. For example, the Moselle-Sure catchment (**Figure 2**) has more than 50% of its area classified into a single geological class (and consequently a single permeability), leading to a uniform geology that limits its





explanatory power. The Vienne basin (**Figure 5**) shows a sudden shift in the geological classes pattern, resulting in sub-catchments with largely homogeneous geology. Here, the predictor variability is highly discrete, with values concentrated at either 0% or 100%. Finally, in the Cinca basin (**Figure 5**), despite abrupt geological shifts from upstream to downstream, the
global map provided higher correlations than the continental map due to its higher geological heterogeneity. As discussed in Section 5.1, the global map has more detailed contours, despite having fewer classes overall, which in this case made it more informative.

Conversely, in regions with high geological heterogeneity, strong correlations between hydrological signatures and geological attributes were observed. This was evident in the Garonne, Narew and Moselle basins (**Figure 5**), and in four out of five
Moselle sub-catchments (**Figure 2**). These findings suggest that spatial gradients in geology are critical for identifying meaningful correlations.

Finally, our findings suggest that the regional maps address the shortcoming of the global (insufficient diversification between classes) and continental (imprecise boundary contours), as they incorporate both higher geological heterogeneity and finer-scale geological details (**Figure 2**). This pattern led to the high hydrological correlation found in the Moselle, for instance
(**Figure 6**).

### 5.3.  Using geological information effectively

### 5.3.1. Reclassification of geological rock formations

Translating maps into meaningful attributes is a challenging task, particularly when these maps contain many classes, with no direct hydrological relevance (Floriancic et al., 2022; Karlsen et al., 2016; Tarasova et al., 2024). This is particularly the case
of geology maps, which often feature a vast array of lithological units. Previous studies have primarily used the percentage coverage of individual geology rock type categories over the catchment area as landscape attributes (Addor et al., 2018; Kratzert et al., 2019; Kuentz et al., 2017). However, in large catchment studies, this approach often results in categories with 0% representation in certain catchments (e.g., carbonate sedimentary rocks), making it difficult to derive meaningful correlations.

We argue that reclassifying geological units into hydrologically relevant categories, such as relative permeabilities, is an essential prerequisite to the identification of meaningful correlations. A comparable effort is the Hydrology of Soil Types (HOST) classification developed for the United Kingdom (Boorman et al., 1995), which categorizes soils based on their influence on hydrological processes. It is also important to reduce the classes dimensionality (e.g., from 31 classes to 4, as in our study). In this way, we ensure a smooth variability of attribute values across catchment, facilitating the determination and
interpretation of correlations. Such approaches are well-established in regional hydrogeological studies (Freeze and Cherry, 1979), where simplifying numerous classes into meaningful hydrological landscape units with similar behaviors facilitates analysis.

In this study, we adopted an approach for reclassifying geological categories aligning with the findings of Fenicia and McDonnell (2022). Yet, we acknowledge that our reclassification still contains subjective choices, and therefore if a different
approach is employed, the users might find different conclusions. It is important though to back such choices according to literature and previous studies (e.g., group rock-types with generally similar permeability together) to ensure that the found correlations mirror as well as possible real causalities.

### 5.3.2. Example of limited information in global maps

Global geology maps do not always contain sufficient information to disentangle different hydrological behaviour. For
instance, while investigating the main differences in geological maps for the Moselle basin (**Figure 2**), we identified that these



discrepancies primarily result from insufficient detail in the classification of siliciclastic rock formations in the first level of the GLiM geological map (**Appendix A**). Siliciclastic rocks encompass formations as sandstone, mudstone and greywacke, but also include shale, rocks with some degree of metamorphic alteration. Despite the significant differences in permeability among these rock types, they are grouped into a single category at this level of classification. The continental and regional
map on the other hand clearly distinct a shale class in their level of classification available.

In our reclassification, we categorized siliciclastic rocks within the high-permeability category, primarily due to the predominance of sandstone and conglomerate formations. This distinction is reflected in the patterns observed in the Moselle and highlights the limitations of relying on a single class to represent rock types with such diverse properties. This lack of informativeness should be carefully considered in LSH studies.

Notably, information on shale within siliciclastic rock formations is often available at the third level of detail in the GLiM dataset (Hartmann et al., 2012). However, to our knowledge, current LSH studies that use this dataset have only utilized the first level of information (Addor et al., 2017; Höge et al., 2023; Klingler et al., 2021; Kratzert et al., 2022; do Nascimento et al., 2024a; Wu et al., 2021). We therefore recommend that future users of the GLiM dataset incorporate the third level of detail when deriving geological attributes for LSH studies to ensure a more nuanced representation of rock formations.

**5.4. The uniqueness of place**

To what extent the ability to regionalize collides with "uniqueness of place" (Beven, 2000) is an open question. When comparing individual catchments, we generally observed varying patterns and controls on streamflow signatures. Large-scale analysis revealed significant differences in the dominant factors influencing streamflow, as well as in the relative importance of climate and landscape controls. Even at the sub-catchment level of the Moselle, the various catchments demonstrated visible
differences in local controls (**Figure 7** and **Figure 8**).

Interestingly, we noted that, in the Moselle basin, increasing map detail appeared to reduce the apparent "uniqueness of place". The regional map presented a sign-consistent correlations between the high-permeability attribute and baseflow. However, there is still considerable remaining variability, underscoring the need for multifactorial approaches to regionalize hydrological processes, ensuring that spatial heterogeneity in catchment attributes is adequately represented.

On the large scale, we also noticed difficulties in devising generalized relationships. Specifically, regardless of whether geological attributes were derived from the global or continental maps, correlations to streamflow signatures, when computed to the whole set of 4,469 sub-catchments over Europe, showed some improvement but remained low (**Appendix C**). This suggests the challenge of treating catchments uniformly across vast areas, reinforcing the importance of regional clustering before aggregation.

Therefore, we recommend that LSH studies aiming either to model rainfall-runoff or to predict streamflow characteristics to prioritize methodologies that incorporate regionalization or clustering approaches, especially when working at continental level. Such approaches are already incorporated in recent machine learning developments for rainfall-runoff prediction using Long Short-Term Memory (LSTM) models, for instance (Kratzert et al., 2019; Nearing et al., 2024). Additionally, studies that achieved strong prediction performance without explicit regionalization, such as in Addor et al. (2018) likely benefited from
the robustness of the methods used in capturing complex relationships and potentially account for spatial heterogeneity that linear or Spearman correlations may miss.



## 6. Conclusions

This study analyzes the correlations between streamflow signatures with climatic and landscape attributes at multiple scales, focusing on a large-scale comparison of 63 river basins, an intermediate-scale analysis of the Moselle basin, and a small-scale experiment involving five Moselle sub-catchments. Each basin contained a variable number of nested catchments, enabling the examination of spatial patterns in streamflow signatures. A particular emphasis was placed on comparing three geology maps of varying levels of detail: global, continental, and regional. Our main conclusions are as follows:

1. **Large-scale analysis**: The analysis revealed distinct controls and rankings of streamflow signatures for each basin. While high correlations were generally observed, no consistent pattern emerged across all basins. Notably, landscape attributes were more frequently identified as dominant baseflow-related signatures than climate attributes. When comparing the global and continental geology maps, the continental map typically provided higher correlations on average, aligning to our initial hypothesis. We attribute this to the complementary strengths and weaknesses of the maps: the global map offers more precise contours for area coverage estimation (higher spatial resolution), while the continental map features more classes, facilitating better diversification in permeability categories.

2. **Intermediate-scale analysis**: This analysis demonstrated how using geology maps of varying detail can lead to drastically different conclusions regarding the dominance of certain landscape attributes in controlling streamflow. The role of geology in controlling baseflow-related signatures transitioned from insignificant to dominant as we moved from the global to the continental, and then to the regional map, aligning to our initial hypothesis. The regional map effectively addressed both the spatial resolution limitations of the continental map and the class diversification limitations of the global map.

3. **Small-scale analysis**: At this scale, distinct patterns were observed between the Moselle sub-catchments. Similar to the large-scale analysis, no single control dominated signature variability across all 5 catchments. However, the regional map was the only one that provided consistent and relatively high correlations across all sub-catchments. It also aligned with our initial, physically motivated two hypotheses about the role of geology in baseflow variability.

Overall, this study highlights the substantial variability in catchment behavior, which may reflect the "uniqueness of place." In particular, there is considerable variability in the extent to which geology influences baseflow-related signatures. However, this variability appears to diminish when higher-quality geological data is used. This suggests that the quality of landscape information, particularly geology, can significantly impact the outcomes of large-scale studies and subsequent interpretations.

**Acknowledgments**

This project was funded by a "Money Follows Cooperation" project (Project No. OCENW.M.21.230) between the Netherlands Organization for Scientific Research (NWO) and the Swiss National Science Foundation (SNSF). This work was further supported by the TU Delft Climate Action Research and Education seed funds.

**Code and data availability**

The used version of the EStreams dataset (v1.2) is stored online at a Zenodo repository (https://doi.org/10.5281/zenodo.14778580) and detailed described by do Nascimento et al. (2024b). Regional geology catchment attributes for the Moselle are available online (https://doi.org/10.5281/zenodo.14779451). All code used in this study is available online at a GitHub repository (https://github.com/thiagovmdon/LSH-quality_geology).



**Author contributions**

FF had the original idea and worked with TN to develop the conceptualization and methodology of the study. TN, JR and FF worked on the data curation. TN wrote all codes used and conducted all formal analysis. FF, MH, and SG supervised the work. The visualizations and original draft of the manuscript were prepared by TN. All co-authors contributed to the review and editing of the manuscript. Funding was acquired by FF and MH. All authors have read and agreed to the current version of the paper.

**Competing of interest**

Some co-authors are members of the editorial board of Hydrology and Earth System Sciences.

**Appendix A: Spatial distribution of the raw geology classes for the five European basins used**

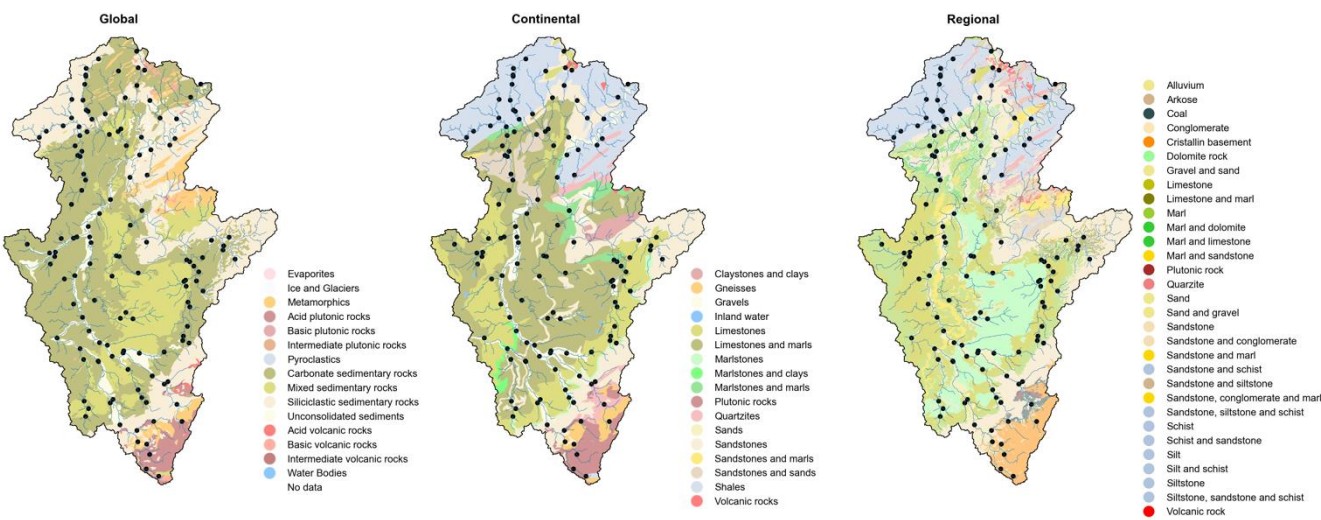

**Figure A 1: Spatial distribution of the original geology classes from the global, continental and regional sources used for the Moselle basin. The colours might represent different categories among the three sources.**





Figure A 2: Spatial distribution of the raw geology classes from the global and continental sources used for the Cinca, Garonne, Viene and Narew basins. The colors might represent different categories among the three sources.



## Appendix B: Size dependency correlations between three selected streamflow signatures and high permeability percentages

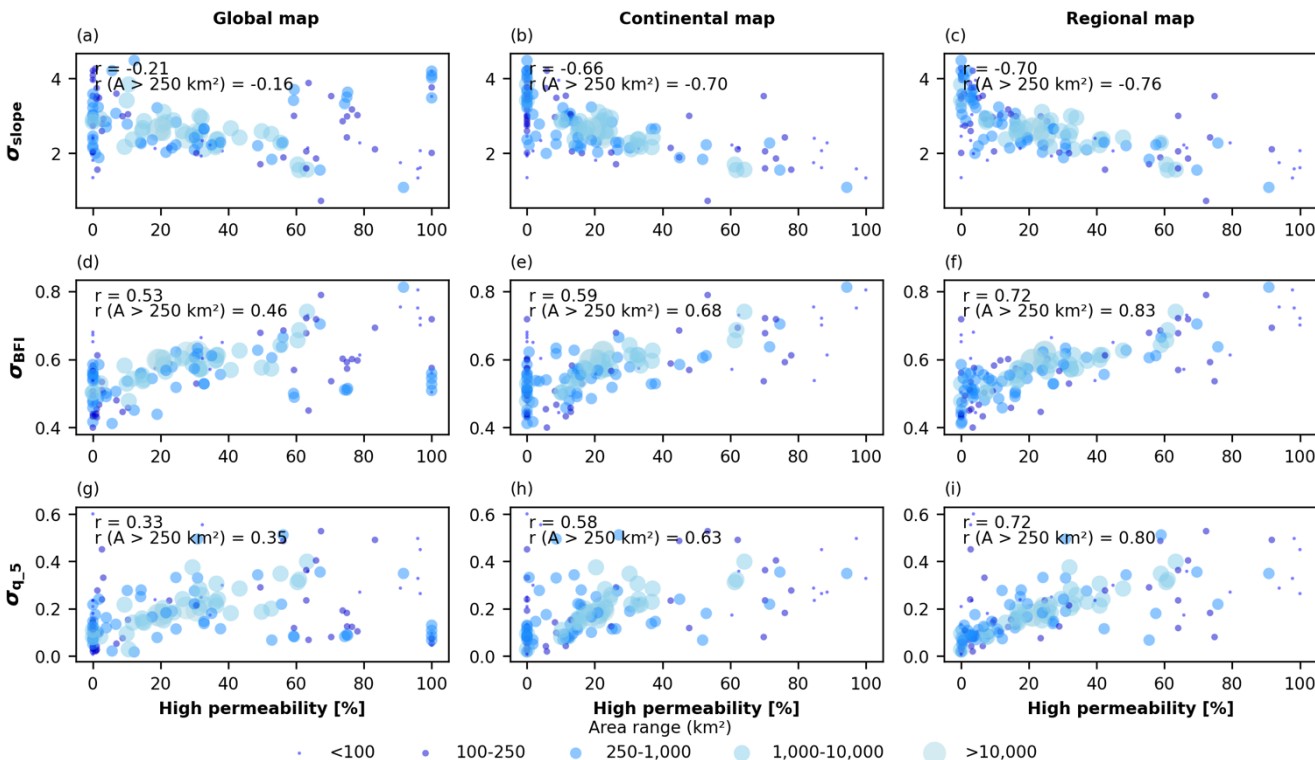

**Figure B 1: Scatter plots showing the correlation between the high-permeability percentages of area in each Moselle sub-catchment versus three selected streamflow signatures. The colours of the circles and their respective sizes represent their area range, varying from below 100 km² to above 10,000 km².**

## Appendix C: Different correlations for baseflow using the European catchments altogether

| Permeability class | Global map ($r_s$) | Continental map ($r_s$) |
|---|---|---|
| Low | 0.02 | 0.03 |
| Medium-low | 0.14 | 0.13 |
| Medium-high | -0.08 | -0.10 |
| High | -0.03 | 0.13 |





**Table C 1: Correlation values ($r_s$) computed using the 4,469 European sub-catchments altogether considering the catchment attributes derived from the global and the continental maps.**

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
