# Peer review of "How do geological map details influence geology-streamflow relationships in large-sample hydrology studies?"

_EGUsphere, 2025_

## Author Comment (AC1)

**Response to Referee 1**

To facilitate the review process, all referee comments are in **black** while the authors response in **blue**.

This paper presents an interesting study demonstrating the value of geology maps in enhancing hydrological understanding. The authors have developed a reclassification method that transforms the original geology map into numerical metrics related to hydrology, and they illustrate the added value of more detailed, small-scale maps. The findings in this paper are valuable for more effectively utilizing geology maps to improve hydrological insights, making it worthy of publication. However, I would like to raise several major and minor concerns that should be addressed prior to publication.

We thank the reviewer for dedicating their time and expertise to review our manuscript. We have carefully addressed each point in detail and hope our responses can make the manuscript in a better shape for publication.

Major Concerns

1. It is unclear why the analysis incorporating climate and landscape attributes is conducted. This study focuses primarily on the value of geology map details, and most of the results are related to the geology map analysis. The analysis using catchment attributes appears to contribute little to the main objective and conclusions of the study. The authors should consider explaining in greater detail how this part of the analysis connects to the study's primary conclusions.

We appreciate the referee's comment and thank them for highlighting this point. We agree that the inclusion of climate and landscape attributes alongside geological maps was not well justified in our original manuscript. Our inclusion was based on two main reasons:

1. **Establishing a comparison of geology attributes with other commonly used attributes:** One of our research aims is assessing the benefit of using geology attributes (from global, continental, and regional maps) along with catchment attributes more commonly used in large-sample hydrology (LSH) studies—i.e., climate, topography, soils, and land use to explain streamflow signatures. By incorporating all other attributes, we could assess if geology-derived variables add explanatory power beyond what is already captured by these commonly used attributes. For example, as shown in **Section 4.1.1** (**Figure 3**), geology attributes presented the highest correlations with baseflow signatures, in comparison to other attributes in many basins, but this varied across spatial scales and map detail levels. Moreover, this figure allows us to understand whether catchments with low geological correlations also have low correlation to other attributes. For example, perhaps the correlation value found of 0.60, is actually very low compared to other attribute groups. But maybe, 0.60 was the highest correlation among all group of attributes. This is the case for the Moselle basin (DEBU1959), for example.

2. **Testing the "landscape vs. climate" assumption:** As mentioned in the introduction, a recurring finding in LSH is that climate often correlates more with streamflow signatures over landscape. In our study, we were also interested to check whether more detailed geological maps can shift this pattern. Our results showed that when performing a LSH study at the basins level, when geology is represented with sufficient detail and appropriate hydrologically relevant indicators (e.g., relative permeability), landscape attributes—especially geology—can present high correlations. This is discussed explicitly in the introduction (**L34–50**) and further developed in the discussion (**Section 5.1, L495–525**). Including climate and other landscape controls allowed us to demonstrate this shift in explanatory power.

We will therefore modify the paper adding a clear justification for this choice.

2. The authors seem to assume a priori that there is an inherent relationship between geology metrics and hydrological signatures, and that a map producing a higher correlation coefficient (rs) is automatically superior. Although the authors attribute this to "physical understanding," from a physical perspective, hydrological signatures are also influenced by climate and land use factors. More detailed information on climate, land use, soil, and topography would also be helpful for interpreting hydrological processes. I suggest that the authors clarify this issue, explain the mechanisms by which geology metrics affect hydrology, and adjust some statements to avoid presuming that geology metrics are the dominant influence.

We thank the referee for this insightful observation. We agree that this causality should not be assumed a priori but should be a result of a deep investigation. We also agree that hydrological signatures result from the interplay of multiple factors—climate, land use, soils, topography, and geology—and that higher correlation values should not be interpreted as implying causality or universal dominance. Our intention is not to claim that geology attributes are inherently more significant, but rather to investigate whether more detailed geological maps enable higher, and more consistent to our stablished hypothesis, correlations at the studied basins. We will revise the manuscript to clarify this and avoid misleading statements. Moreover, we will avoid the use of "superior", "more significant", "better" or misleading statistical terms in the manuscript (also answering your further points). Please see also our response to the **referee 2**.

3. The inherent or fundamental differences among the three maps should be summarized somewhere in Section 2.2. At first glance, the differences appear to be in the spatial range resolution, but this factor does not actually explain the differences observed in the correlation analyses. Clarifying these intrinsic differences would help readers better understand why the regional map performs better.

This is a very good discussion point, and we thank the referee for highlighting it. We therefore will integrate section 2.2. with a clear overview of such differences and add the Table below to summarize them. With that, we believe that the readers do not need to wait to the discussion to see an overview of these differences.

**2.2.4 Summary of main differences in the geological maps**

*The three geological maps used in this study differ not only in spatial scale but also in other fundamental ways, such as the number of lithological classes they include and how finely they delineate geological boundaries. Table X summarizes these differences, which help to interpret why different maps lead to different correlation patterns, as further discussed in Section 5.1 and Section 5.2.*

**Table X.** *Summary of the main differences between the three geological maps*

| Main characteristics | Global | Continental | Regional |
|---|---|---|---|
| Spatial coverage | World | Pan-European | Moselle basin |
| Geological number of classes (spatial heterogeneity) | Low (16 classes) | High (32 classes) | High (32 classes) |
| Geological detail (Boundary precision) | High (detailed contours) | Lower (smoothed boundaries) | High (detailed contours) |

Minor Concerns

L188: Consider mentioning the five selected basins for detailed analysis at this point.
Thank you for this suggestion. We will correct L188 to:

L188: Large scale: This level included 63 river basins across Europe, and selected five basins (the Moselle, Cinca, Garonne, Vienne, and Narew) for further exploration, as illustrated in Figure 1a.

L236: Should "Five" be corrected to "Four"?
Yes, thank you for spotting this detail out. We will correct it to four:
**L236:** "four"

Section 3.4.3: The use of the geology map seems to be missing here.
Thank you for pointing this out. Besides, I think we should correct some sentences since we focus on this scale on the baseflow index signature. Therefore, the section will be corrected to align more or less as follows in the revised manuscript:

**3.4.3. Small-scale**
For the five catchments of the Moselle, we employed the following methodology:

- We calculated the correlation between the baseflow index signature and 47 catchment attributes for each of the five catchments of the Moselle, using all available nested sub-catchments within each catchment. This resulted in a total of 1 x 47 x 5 = 235 correlation values $r_s$.
- Instead of selecting only the maximum $|r_s|$ value per group, we conducted a more refined analysis here, focusing on the coherence of correlations, mainly the derived from geological attributes, consistent with the hypotheses raised in the Introduction. This more in-depth analysis aimed to assess how local factors might alter the relationships identified at broader scales, allowing us to distinguish correlations that are likely to reflect true causal relationships.

L328: A statistical analysis is needed to determine whether the higher rs derived from the continental map compared to the global map is statistically significant. A similar analysis should be conducted elsewhere when describing the differences among the three maps.
Thank you very much for pointing this detail out. Our intention here in the results is to present the results and avoid overselling points. Therefore, we will review the text to avoid the misinterpretation that one is statistically significantly higher than the other. We will also have a look over the entire manuscript to review any other statement with the same nature.

*L328: Comparing the maps of different detail, the continental map appeared with average $|rs| = 0.42$, while the global map presented an average $|rs| = 0.40$.*

Paragraph around L330: In the right part of Figure 3, many basins exhibit much lower rs values with the continental map compared to the global map. We cannot simply regard the higher continental rs in the left part as an "added value" while ignoring the lower values in the right part. This discrepancy reflects the divergence between the two maps, as also shown in Figure 2. The divergence between the maps should be analyzed
carefully, rather than simply judging which map performs better based solely on a higher rs.

We thank the reviewer for this insightful observation. We agree that using higher correlation values alone to judge one map as "better" is overly simplistic, especially given that many basins exhibit lower correlations with the continental map compared to the global map (as seen on the right side of Figure 3). To address this, we will revise the relevant text to avoid framing one map as superior. Instead, we plan to emphasize that the maps provide complementary information—e.g., the continental map offers greater class diversity, while the global map provides finer boundary detail. We believe this adjusted framing will better reflect the divergence between the maps and the complexity of their respective contributions to the analysis.

*L330: This suggests that the continental map offers complementary value in basins where the global map performs poorly, due to its higher geological heterogeneity. However, in some cases, the global map yields higher correlations—likely due to its finer boundary detail—highlighting that both maps carry distinct, context-dependent advantages rather than one being universally superior. More on that is discussed in Section 5.2.*

Paragraph around L480: The conclusion that the regional map provides the most stable correlations appears inappropriate. Among the three regional metrics, only one consistently produces the same sign, while the other two have one and two exceptions, respectively. However, similar patterns can be found in many other metrics.

Thank you for your comment. We agree that the original phrasing may have overstated the consistency of the regional map. While it is true that only one of the three regional geological attributes consistently produced correlations with the same sign across all five catchments, it was also the only map-attribute combination that showed both consistent sign and correlation values above 0.40. We will therefore revise/correct the sentence in the manuscript to reflect this more cautiously and accurately.

*L477-479: Among all geological attributes considered, only the high-permeability attribute from the regional map consistently produced correlations with the same sign (positive) and values above 0.40 across all five catchments.*

Figure 8: Consider adding separating lines between the different groups to enhance the figure's readability.

Thank you for your comment. We will take this into consideration and the figure will be updated as follows:

[Figure]

L483: The value "0.93" is not found in Figure 8.
Thank you for your comment. The value should be "0.90", therefore we will correct the statement.

---

## Author Comment (AC2)

**Response to referee 2**

To facilitate the review process, all referee comments are in **black** while the authors response in **blue**.

This paper provides an interesting assessment of geological information on streamflow signatures. The study of this work at separate scales is potentially interesting and a refreshing perspective for large-sample hydrology studies. The study could be a useful addition to the hydrological literature, but before I can recommend publication, the following points need to be clarified:

We sincerely thank the reviewer for their time to review our manuscript. We greatly appreciate their thoughtful and constructive feedback, which we found to be both valuable and encouraging. We have carefully considered each comment and have provided detailed responses to all of them. We hope that our revisions and clarifications can bring the paper closer to being suitable for publication.

Please note that, due to their closely related nature, the next four comments are addressed together in a combined response.

- "Normal" (typical) large sample hydrology analyses study (statistics) relationships across the entire study domain at once. This paper takes a different approach by studying local (basin-scale) relationships across many basins at once. This is a valuable addition to the literature, but this contrast needs to be better highlighted, as it fundamentally changes the question that is asked (the current paper does not check continental scale dominance of catchment attributes) but the local scale dominance of factors (across an entire continent). This different approach inherently changes the answers one gets, but will also change the question that is asked, and as such this should be better contrasted to the existing literature. Right now this

- In the introduction, the manuscript reflects on why landscape appears to have a small impact on hydrology according to LSH studies. The work states this may be due to several reasons but maybe overlooks (according to my gut feeling, not that I have no formal evidence) the most obvious reason: the landscape can have a very important role on hydrology but how important this role is depends on the diversity of climates considered. Most LSH hydrology studies are across tremendous climate gradients (e.g. the current continental-scale study) and this easily dominates the role of many (but not all) hydrological signatures. Sure, landscape will (strongly) modify how incoming precipitation is partitioned, but, if climate differences are big enough, it cannot override these climate effects. And if we are not yet super successful at normalizing for climate effects, effectively learning about landscape in LSH studies across large geographical domains becomes challenging. If LSH were conducted with many catchments in a similar climate setting, the role of landscape would become (relatively seen) much more important and very likely easier (but still challenging, due to reasons you also state) to identify.

- The approach that is used already seems to partly acknowledge this, as at large scales, it checks for correlations at the scale of a basin, which reduces climate gradients. Explain this. But also explain thereby that the results you get are "dominant controls" on the scale that you study, and not at the continental scale when the large scales is studied.

- In the discussion, the paper also reflects on this by comparing it to previous studies "(Addor et al., 2018; Beck et al., 2015; Kratzert et al., 2019; Kuentz et al., 2017) that found that climate is a stronger control on signatures than landscape. The current manuscript needs to be more precise in stating that "if the scale of the assessed correlations changes, the question changes" (and thus the answer).

We sincerely thank the reviewer for the thoughtful feedback. We especially appreciate the recognition of our alternative approach to LSH, and we fully agree with the need to better highlight how this contrasts with traditional LSH studies. In typical LSH work, correlations are computed across all catchments simultaneously, aiming to identify continental-scale dominant controls. In contrast, our study takes a basin-wise approach, analysing relationships within individual regions/clusters. This shift inherently changes the research question: we are not aiming to assess universal drivers across Europe, but rather to understand what dominates basins in different climates and landscapes.

Additionally, we appreciate the reviewer's suggestion regarding the confounding effect of large climate gradients in LSH. This is a valuable point that supports our argument. We will restructure the introduction stating that climate

heterogeneity across large regions may mask landscape effects—further justifying our approach of analysing catchments grouped by region.

- The choice to use Spearman correlations is OK, but not very convincing. I understand that every method comes with its problems (and advantages) but it would be good if the reader gets more confidence that a method that not only considers these individual correlations (but tests for interaction of more catchment attributes) would come to broadly similar conclusions.

We thank the reviewer for raising this important point. We agree that univariate correlation methods have inherent limitations. Particularly in their inability to detect interactions or nonlinear relationships among predictors. Our primary motivation for using Spearman correlations was to maintain interpretability and transparency when comparing across thousands of catchments and multiple geological datasets. Spearman provides a straightforward basis for comparing the strength of individual attribute-signature relationships across scales.

That said, we fully acknowledge that this approach cannot capture more complex interactions between climate, landscape, and geological variables. To address this, we point the reviewer to Section 3.3, where we explicitly discussed this limitation and pointed to previous studies that employed multivariate or machine learning approaches (e.g., Addor et al., 2018; Beck et al., 2015; Kratzert et al., 2019). This helps reassure the reader that while more advanced methods might improve predictive performance, the broad conclusions about the scale-dependent influence of geology and landscape are likely to hold.

- The use of the "maximum $|r\_s|$" seems somewhat odd when not all groups have an equal amount of catchment predictors. Is there a risk that groups that have more predictors are considered to be more dominant, not because they physically are, but because they are more numerous? (I do not expect this to be dominant, but maybe good to consider).

Thank you very much for your comment. There are two key points we would like to clarify:

1. **Group size:** For the geology groups—global, continental, and regional—each contains exactly four variables, corresponding to the four permeability classes. For climate, soils, land use, and topography, the number of variables per group does vary (as shown in Table 3), but the differences are not extreme.

2. **Consequent choice of maximum:** We chose the maximum $|r_s|$ to minimize such differences in group sizes. Our goal was to highlight the highest relationships for each group. Using the median or mean could underrepresent the influence of a group that contains one or two strong, but informative, variables—especially for attributes like geology, where relevant effects may be subtle but specific.

Therefore, we will clarify Section 3.4.1 to acknowledge this trade-off and clarify that while group size may introduce some bias, we judged the maximum $|r_s|$ to be the most informative and interpretable summary metric for this study's purpose.

- Correlation of catchment attributes and signatures seem to be interpreted as "correlation = causation", but that is very speculative. I understand that this often happens in LSH, but here the approach is rather with a scattergun approach: the work studies a very long list of correlations (without physical hypotheses how various factors shape individual catchment attributes) and then picks the strongest correlation at the catchment scale. This seems sensitive to spurious correlations.

Thank you very much for this important comment. It is not our intention to equate correlation with causation, and we fully acknowledge that statistical correlations—especially in large-sample studies—can reflect associations without implying a direct physical mechanism. We also agree that some statements in the original manuscript may have unintentionally given the impression that strong correlations were interpreted as causal relationships. We have carefully reviewed the full text to revise such language and ensure that findings are framed appropriately in terms of statistical association rather than physical inference.

We will also clarify that:

- Spearman correlations are used to highlight potential dominant relationships worth further investigation—not to claim definitive causal links.
- Interpretation of correlations should be made cautiously and considering hydrological expectations and previous literature.

Specific changes will include:

- Rewording sentences in the **Abstract**, **Introduction**, and **Discussion** to avoid causal language (e.g., "control," "impact", "causal") where correlation is meant. Moreover, reword "stronger" to "higher" when discussing correlation values.
- Adding clarifying statements in Section 3.3 and 5.4 that reinforce the distinction between correlation and causation.

Therefore, we believe that with such corrections (see also comments from referee 1), this correlation/causality misinterpretation will be solved, and the manuscript will have a better shape for publication.

- FIg 3: I think it would help if this information was also shown in a different way. For example, scatter plots between geology global values and the other categories. In addition, it would be very helpful it is shown how strongly the best predictors of the different classes are correlated with another (again with scatter plots)

Thank you very much for this constructive suggestion. We agree that comparing geology predictors directly and assessing inter-correlations across attribute groups could be valuable for exploring redundancy or coherence among predictors.

However, after consideration, we believe that including additional scatter plots for all geology predictors (global vs. continental vs. regional) and cross-correlations between the best predictors of each group would introduce significant complexity without proportionate interpretive gain. Since many of the predictors are categorical or derived from reclassifications (e.g., permeability fractions), their relationships are often non-linear, bounded, or collinear by design (summing to 1), which makes such scatter plots potentially misleading or difficult to interpret without heavy qualification.

Moreover, part of the strength of our current approach—especially in Figure 3—is its ability to condense a large volume of correlation results across multiple basins and attribute groups into a format that is both interpretable and comparative. Adding multi-dimensional scatter plots would complicate this clarity and risk distracting from the paper's core objective: understanding how dominant controls vary with scale and geological map detail.

For these reasons, we have chosen not to add the suggested scatter plots, though we acknowledge the value of such diagnostics and may explore them in a follow-up analysis or supplementary study focused more explicitly on predictor redundancy and interaction.

- Fig. 4: this comparison between the effects (or correlations) of global geology vs continental geology is useful, but it would benifit from shwong a but more than just the data points and 1:1 line. What are the average values of both (you can show the average X and Y coordiante, and this summarizes how well one predicts vs the other, and how well they overall predict). You can also show the correlation (coefficient) between the two data points to show how strongly related they are to another. Such things could greatly help the interpretation of these plots.

Thank you very much for this comment. These are extremely valuable insights into the figure. We took them into consideration and here you may find an updated version of it.

[Figure]

**Legend**

● Cinca    ● Garonne    ● Vienne    ● Moselle    ● Narew    ● Other basins    --- 1:1 line    — Mean

---

## Author Response (AR1)

To facilitate the review process, all referee comments are in **black** while the authors response in **blue**.

**Final response**

**Public justification (visible to the public if the article is accepted and published)**:

Dear Authors,

Thank you for submitting your manuscript. Based on the evaluations from both reviewers, we find that your study presents a valuable and original contribution to the field by exploring the hydrological significance of geology maps and applying a basin-scale analysis across a continental domain. However, several major issues must be addressed before the paper can be considered for publication. These include clarifying the role of catchment attributes in your analysis, justifying methodological choices such as the use of Spearman correlations and "maximum $|r\_s|$", and clearly distinguishing your study's approach from traditional large-sample hydrology frameworks. Both reviewers also highlight the need for improved explanation of the conceptual underpinnings, more rigorous interpretation of correlations, and enhancements to the visual presentation of results. We therefore invite you to revise your manuscript accordingly and resubmit for further consideration.

Best regards,
Fuqiang Tian
Tsinghua University

We thank the editor and reviewers for dedicating their time for and expertise to review our manuscript. We have carefully addressed each point in detail and updated the manuscript accordingly. We hope that the current version of the manuscript is now in shape for publication. The response is organized below addressing each of the points mentioned, with specific details regarding referee comments.

This paper presents an interesting study demonstrating the value of geology maps in enhancing hydrological understanding. The authors have developed a reclassification method that transforms the original geology map into numerical metrics related to hydrology, and they illustrate the added value of more detailed, small-scale maps. The findings in this paper are valuable for more effectively utilizing geology maps to improve hydrological insights, making it worthy of publication. However, I would like to raise several major and minor concerns that should be addressed prior to publication.

We thank the reviewer for dedicating their time for and expertise to review our manuscript. We have carefully addressed each point in detail below.

Major Concerns

1. It is unclear why the analysis incorporating climate and landscape attributes is conducted. This study focuses primarily on the value of geology map details, and most of the results are related to the geology map analysis. The analysis using catchment attributes appears to contribute little to the main objective and conclusions of the study. The authors should consider explaining in greater detail how this part of the analysis connects to the study's primary conclusions.

We appreciate the referee's comment and thank them for highlighting this point. We agree that the inclusion of climate and landscape attributes alongside geological maps was not well justified in our original manuscript. Our inclusion was based on two main reasons:

1. **Establishing a comparison of geology attributes with other commonly used attributes:** One of our research aims is assessing the benefit of using geology attributes (from global, continental, and regional maps) along with catchment attributes more commonly used in large-sample hydrology (LSH) studies— i.e., climate, topography, soils, and land use to explain streamflow signatures. By incorporating all other attributes, we could assess if geology-derived variables add explanatory power beyond what is already captured by these commonly used attributes. For example, as shown in **Section 4.1.1** (**Figure 3**), geology attributes presented the highest correlations with baseflow signatures, in comparison to other attributes in many basins, but this varied across spatial scales and map detail levels. Moreover, this figure allows us to understand whether catchments with low geological correlations also have low correlation to other attributes. For example, perhaps the correlation value found of 0.60, is actually very low compared to other attribute groups. But maybe, 0.60 was the highest correlation among all group of attributes. This is the case for the Moselle basin (DEBU1959), for example.

2. **Testing the "landscape vs. climate" assumption:** As mentioned in the introduction, a recurring finding in LSH is that climate often correlates more with streamflow signatures over landscape. In our study, we were also interested to check whether more detailed geological maps can shift this pattern. Our results showed that when performing a LSH study at the basins level, when geology is represented with sufficient detail and appropriate hydrologically relevant indicators (e.g., relative permeability), landscape attributes—especially geology—can present high correlations. This is discussed explicitly in the introduction (**L34–50**) and further developed in the discussion (**Section 5.1, L495–525**). Including climate and other landscape controls allowed us to demonstrate this shift in explanatory power.

We have modified the paper adding a clear justification for this choice in the **Introduction** and **Methods** section:

*L101-102: "While the primary aim was to evaluate the role of geological detail, climatic and other landscape attributes were also included for comparative purposes."*

*L250-254: "Although geology is the primary focus of this study, we included other landscape and climate attributes to provide a comparative baseline. This allows us to assess whether improvements in geological data also lead to stronger explanatory power relative to other domains and helps place our findings in the context of broader hydrological understanding. For example, comparing across attribute groups enables us to interpret*

*whether a correlation of 0.60 for a geology variable is relatively strong or weak compared to topography, soils, land use, or climate attributes."*

2. The authors seem to assume a priori that there is an inherent relationship between geology metrics and hydrological signatures, and that a map producing a higher correlation coefficient (rs) is automatically superior. Although the authors attribute this to "physical understanding," from a physical perspective, hydrological signatures are also influenced by climate and land use factors. More detailed information on climate, land use, soil, and topography would also be helpful for interpreting hydrological processes. I suggest that the authors clarify this issue, explain the mechanisms by which geology metrics affect hydrology, and adjust some statements to avoid presuming that geology metrics are the dominant influence.

We thank the referee for this insightful observation. We agree that this causality should not be assumed a priori but should be a result of an investigation. We also agree that hydrological signatures result from the interplay of multiple factors—climate, land use, soils, topography, and geology—and that higher correlation values should not be interpreted as implying causality or universal dominance. Our intention is not to claim that geology attributes are inherently more significant, but rather to investigate whether more detailed geological maps enable higher, and more consistent to our stablished hypothesis, correlations at the studied basins. We have revised the manuscript to clarify this and avoid misleading statements. Please see also our response to the **referee 2** with a detailed overview of the modifications.

3. The inherent or fundamental differences among the three maps should be summarized somewhere in Section 2.2. At first glance, the differences appear to be in the spatial range resolution, but this factor does not actually explain the differences observed in the correlation analyses. Clarifying these intrinsic differences would help readers better understand why the regional map performs better.

This is a very good discussion point, and we thank the referee for highlighting it. We therefore have added section 2.2.4 to the text with a clear overview of such differences:

*L181-188:*

*2.2.4 Summary of the main differences in the geological maps*
*The geological maps used in this study differ not only in spatial scale but also in the number of lithological classes they include and how finely they delineate geological boundaries. Notably, the same location may be assigned different lithologies depending on the map used (e.g., **Figure A1** and **Figure A2**), resulting in varying geological classifications for the same catchment. These differences are summarized in **Table 1** and can also be seen in **Figure 2**. While the global map presents a lower number of classes than the continental and regional maps, both the global and regional maps feature more detailed boundary contours.*

*Table 1. Summary of the main differences between the geological maps*

| Main characteristics | Global | Continental | Regional |
|---|---|---|---|
| Spatial coverage | World | Pan-European | Moselle basin |
| Number of geological classes used (spatial heterogeneity) | Low (16 classes) | High (31 classes) | High (31 classes) |
| Boundary delineation | High (detailed contours) | Low (smoothed boundaries) | High (detailed contours) |

Minor Concerns

L188: Consider mentioning the five selected basins for detailed analysis at this point.

We corrected it to:

*L193: "Large scale: This level included 63 river basins across Europe, and selected five basins (the Moselle, Cinca, Garonne, Vienne and Narew) for further exploration, as illustrated in **Figure 1a**."*

L236: Should "Five" be corrected to "Four"?
Yes, thank you for spotting this detail out. We corrected it to four.

Section 3.4.3: The use of the geology map seems to be missing here.
Thank you for pointing this out. The section was corrected as follows:

*L338-347:*

*3.4.3. Small-scale*

*For the five catchments of the Moselle, we employed the following methodology:*

- *We calculated the correlation between the baseflow index and 47 catchment attributes for each of the five catchments of the Moselle, using all available nested sub-catchments within each catchment. This resulted in a total of 1 x 47 x 5 = 235 correlation values rs.*
- *Instead of selecting only the maximum |rs| value per group, we conducted a more refined analysis here, focusing on the coherence of correlations, mainly the ones derived from geological attributes, consistent with the hypotheses raised in the Introduction.*

*This more in-depth analysis aimed to assess how local factors might alter the relationships identified at broader scales, allowing us to distinguish correlations that are likely to reflect hydrological expectation and previous literature.*

L328: A statistical analysis is needed to determine whether the higher rs derived from the continental map compared to the global map is statistically significant. A similar analysis should be conducted elsewhere when describing the differences among the three maps.
Thank you very much for pointing this detail out. Our intention here in the results is to present the results and avoid overselling points. Therefore, we reviewed the text to avoid the misinterpretation that one is statistically significantly higher than the other. We also had a look over the entire manuscript to review any other statement with the same nature. See also the response to **referee 2**.

*L360-361: "Comparing the geological information from maps of different detail, the continental map appeared with mean |rs| = 0.42, minimum 0.05 and maximum 0.87, while the global map presented a mean |rs| = 0.40, minimum 0.01 and maximum 0.80."*

Paragraph around L330: In the right part of Figure 3, many basins exhibit much lower rs values with the continental map compared to the global map. We cannot simply regard the higher continental rs in the left part as an "added value" while ignoring the lower values in the right part. This discrepancy reflects the divergence between the two maps, as also shown in Figure 2. The divergence between the maps should be analyzed
carefully, rather than simply judging which map performs better based solely on a higher rs.

We thank the reviewer for this insightful observation. We agree that using higher correlation values alone to judge one map as "better" is overly simplistic, especially given that many basins exhibit lower correlations with the continental map compared to the global map (as seen on the right side of Figure 3). To address this, we revised the relevant text to avoid framing one map as superior. Instead, we now emphasize that the maps provide complementary information—e.g., the continental map offers greater class diversity, while the global map provides finer boundary detail. We believe this adjusted framing better reflects the divergence between the maps and the complexity of their respective contributions to the analysis.

*L363-367: "This suggests that the continental map offers complementary value in basins where the global map performs poorly, probably due to its higher geological heterogeneity. However, in some cases, the global map*

*yields higher correlations—likely due to its finer boundary detail—highlighting that both maps carry distinct, context-dependent advantages (Table 1). More on that is discussed in Section 5.2.”*

Paragraph around L480: The conclusion that the regional map provides the most stable correlations appears inappropriate. Among the three regional metrics, only one consistently produces the same sign, while the other two have one and two exceptions, respectively. However, similar patterns can be found in many other metrics.

Thank you for your comment. We agree that the original phrasing may have overstated the consistency of the regional map. While it is true that only one of the three regional geological attributes consistently produced correlations with the same sign across all five catchments, it was also the only map-attribute combination that showed both consistent sign and correlation values above 0.40. We have therefore revised/corrected the sentence in the manuscript to reflect this more cautiously and accurately.

*L514-515: “Among all geological attributes considered, only the high-permeability attribute from the regional map consistently produced correlations with the same sign (positive) and values above 0.40 across all five catchments”*

Figure 8: Consider adding separating lines between the different groups to enhance the figure's readability.

Thank you for your comment. We will take this into consideration and the figure has been updated.

[Figure]

L483: The value "0.93" is not found in Figure 8.
Thank you for your comment. The value should be "0.90", therefore we corrected it.

**Response to referee 2**

To facilitate the review process, all referee comments are in **black** while the authors response in **blue**.

This paper provides an interesting assessment of geological information on streamflow signatures. The study of this work at separate scales is potentially interesting and a refreshing perspective for large-sample hydrology studies. The study could be a useful addition to the hydrological literature, but before I can recommend publication, the following points need to be clarified:

We sincerely thank the reviewer for their time to review our manuscript. We greatly appreciate their thoughtful and constructive feedback, which we found to be both valuable and encouraging. We have carefully considered each comment and have provided detailed responses to all of them.

Please note that, due to their closely related nature, the next four comments are addressed together in a combined response.

- "Normal" (typical) large sample hydrology analyses study (statistics) relationships across the entire study domain at once. This paper takes a different approach by studying local (basin-scale) relationships across many basins at once. This is a valuable addition to the literature, but this contrast needs to be better highlighted, as it fundamentally changes the question that is asked (the current paper does not check continental scale dominance of catchment attributes) but the local scale dominance of factors (across an entire continent). This different approach inherently changes the answers one gets, but will also change the question that is asked, and as such this should be better contrasted to the existing literature. Right now this

- In the introduction, the manuscript reflects on why landscape appears to have a small impact on hydrology according to LSH studies. The work states this may be due to several reasons but maybe overlooks (according to my gut feeling, not that I have no formal evidence) the most obvious reason: the landscape can have a very important role on hydrology but how important this role is depends on the diversity of climates considered. Most LSH hydrology studies are across tremendous climate gradients (e.g. the current continental-scale study) and this easily dominates the role of many (but not all) hydrological signatures. Sure, landscape will (strongly) modify how incoming precipitation is partitioned, but, if climate differences are big enough, it cannot override these climate effects. And if we are not yet super successful at normalizing for climate effects, effectively learning about landscape in LSH studies across large geographical domains becomes challenging. If LSH were conducted with many catchments in a similar climate setting, the role of landscape would become (relatively seen) much more important and very likely easier (but still challenging, due to reasons you also state) to identify.

- The approach that is used already seems to partly acknowledge this, as at large scales, it checks for correlations at the scale of a basin, which reduces climate gradients. Explain this. But also explain thereby that the results you get are "dominant controls" on the scale that you study, and not at the continental scale when the large scales is studied.

- In the discussion, the paper also reflects on this by comparing it to previous studies "(Addor et al., 2018; Beck et al., 2015; Kratzert et al., 2019; Kuentz et al., 2017) that found that climate is a stronger control on signatures than landscape. The current manuscript needs to be more precise in stating that "if the scale of the assessed correlations changes, the question changes" (and thus the answer).

We sincerely thank the reviewer for the thoughtful feedback. We especially appreciate the recognition of our alternative approach to LSH, and we fully agree with the need to better highlight how this contrasts with traditional LSH studies. In typical LSH work, correlations are computed across all catchments simultaneously, aiming to identify continental-scale dominant controls. In contrast, our study takes a basin-wise approach, analysing relationships within individual regions/clusters. This shift inherently changes the research question: we are not aiming to assess universal drivers across Europe, but rather to understand what dominates basins in different climates and landscapes. Additionally, we appreciate the reviewer's suggestion regarding the confounding effect of large climate gradients in LSH. This is a valuable point that supports our argument.

We have restructured the manuscript stating that climate heterogeneity across large regions may mask landscape effects—further justifying our approach of analysing catchments grouped by region:

*L14-15:* *"To distinguish landscape influences from the otherwise dominant influence of climate, we conducted separate analyses on nested basins."*

*L55-60:* *"Moreover, it is well known that dominant hydrological processes, as well as their controls, change with scale (e.g., Blöschl and Sivapalan, 1995). At very large scales, climate is often dominant because it constrains the water and energy supply and regulates whether precipitation falls as rain or snow (Budyko and Miller, 1974; Knoben et al., 2018). This may, at least to some extent, override the role of landscape attributes (e.g., geology) in controlling the streamflow response (Gnann et al., 2019). In this case, it becomes important to filter out climatic influences, for instance by studying sub-domains separately or by using metrics that integrate climatic and landscape factors (Gleeson et al., 2011a; van Oorschot et al., 2024)."*

*L330-334:* *"Furthermore, while most large-sample hydrology (LSH) studies compute correlations across all catchments simultaneously, our approach differs by analyzing 63 individual river basins separately across a continental domain. This framing limits the influence of broad climate gradients, which can mask landscape effects. Therefore, by focusing on climatically consistent units, i.e., individual river basins with nested catchments, we reduce climate variability and improve our ability to detect basin-specific controls on streamflow signatures."*

*L530-534:* *"While earlier studies typically examined large-sample datasets collectively, highlighting the dominant role of climate, we analyzed each basin individually to better capture the influence of landscape characteristics. Second, our findings underscore the challenge of generalizing landscape influences on hydrological signatures. The large-scale analysis reveals substantial variability in how landscape attributes relate to streamflow signatures."*

*L663-666:* *"Lastly, the use of river basins as the primary analysis unit at the large-scale proved to be a particularly effective approach. Unlike the more common approach of aggregating all data, this method enabled us to isolate and interpret streamflow controls in a way that would likely have been obscured at broader scales."*

- The choice to use Spearman correlations is OK, but not very convincing. I understand that every method comes with its problems (and advantages) but it would be good if the reader gets more confidence that a method that not only considers these individual correlations (but tests for interaction of more catchment attributes) would come to broadly similar conclusions.

We thank the reviewer for raising this important point. We agree that univariate correlation methods have inherent limitations. Particularly in their inability to detect interactions or nonlinear relationships among predictors. Our primary motivation for using Spearman correlations was to maintain interpretability and transparency when comparing across thousands of catchments and multiple geological datasets. Spearman provides a straightforward basis for comparing the strength of individual attribute-signature relationships across scales.

That said, we fully acknowledge that this approach cannot capture more complex interactions between climate, landscape, and geological variables. To address this, we have improved section 3.3 and hope that this helps reassure the reader that while more advanced methods might improve predictive performance, the broad conclusions about the scale-dependent influence of geology and landscape are likely to hold.

*L305-313:* *"While this univariate approach is useful for interpretability and isolating the influence of individual attributes, it has key limitations. First, it does not capture interactions among multiple variables or account for multicollinearity—an important consideration given that many climate and landscape attributes are interdependent (Mathai and Mujumdar, 2019). Moreover, univariate correlations may be prone to spurious relationships, especially in complex hydrological systems.*

*Although more advanced statistical and machine learning methods—such as multiple regression, random forests, or other multivariate approaches—could better disentangle the individual effects of geology and control for confounding variables (Addor et al., 2018; Beck et al., 2015; Kuentz et al., 2017), the goal of this study is not to maximize predictive accuracy. Rather, our focus is on interpretability, physical consistency, and evaluating how geological map detail influences our understanding of streamflow generation mechanisms."*

- The use of the "maximum $|r\_s|$" seems somewhat odd when not all groups have an equal amount of catchment predictors. Is there a risk that groups that have more predictors are considered to be more

dominant, not because they physically are, but because they are more numerous? (I do not expect this to be dominant, but maybe good to consider).

Thank you very much for your comment. We have included the following sentences in our manuscript:

**L325-329:** *"We selected the maximum rather than the mean or median $|r_s|$ to highlight the strongest attribute–signature relationships within each group. While we acknowledge that groups with more variables may be more likely to yield higher maximum values, the goal here was not to compare groups statistically, but to identify potential dominant controls. Mean or median values could obscure key relationships by averaging over non-informative variables, whereas the maximum highlights potentially meaningful signals worthy of further investigation."*

- Correlation of catchment attributes and signatures seem to be interpreted as "correlation = causation", but that is very speculative. I understand that this often happens in LSH, but here the approach is rather with a scattergun approach: the work studies a very long list of correlations (without physical hypotheses how various factors shape individual catchment attributes) and then picks the strongest correlation at the catchment scale. This seems sensitive to spurious correlations.

Thank you very much for this important comment. It is not our intention to equate correlation with causation, and we fully acknowledge that statistical correlations—especially in large-sample studies—can reflect associations without implying a direct physical mechanism. We also agree that some statements in the original manuscript may have unintentionally given the impression that strong correlations were interpreted as causal relationships. We have carefully reviewed the full text to revise such language and ensure that findings are framed appropriately in terms of statistical association rather than physical inference.

**General changes:** We have revised the full manuscript and changed any statement using "better" or "worse" to "higher" or "lower". Moreover, we also deleted any causal language (e.g., "control," "impact", "causal") where correlation is meant. Non-exhaustive examples include:

**L433:** *"(…) a slightly **higher** correlation (…)"*

**L489:** *"(…) show **higher** geological correlation (…)"*

**L511:** *"(…) reached a much **higher** one to (…)"*

**L513:** *"(…) indicates a **lower** correlation to medium-high permeability (…)"*

**L631:** *"(…) the **highest** correlated attributes groups (…)"*

**Title**: We also adjusted the title to:

*"How do geological map details influence the identification of geology-streamflow relationships in large-sample hydrology studies?"*

- FIg 3: I think it would help if this information was also shown in a different way. For example, scatter plots between geology global values and the other categories. In addition, it would be very helpful it is shown how strongly the best predictors of the different classes are correlated with another (again with scatter plots)

Thank you very much for this constructive suggestion. We agree that comparing geology predictors directly and assessing inter-correlations across attribute groups could be valuable for exploring redundancy or coherence among predictors. Due to the already manuscript length, we have added two figures based on the referee suggestion to Appendix, but have commented them in the manuscript:

**L374-385: "***Among these six climatic and landscape groups, the attributes with most often the highest correlation to baseflow index were the snow cover mean (climate), the fraction of steep area (topography), the mean NDVI (land use), the mean fraction of silt (soils), the high-permeability class (global geological map) and the low-permeability (continental). The relationship among these attributes and baseflow index are depicted*

*with scatter plots in Figure D 1, and between these attributes themselves in Figure D 2. The first plot gives an overview of the relationships underlying **Figure 3**, and the second helps to give some indication how the different catchment attributes are correlated with each other.*

*In fact, only correlation between baseflow index and mean NDVI (Figure D 1) reached a value above 0.30 when evaluating all catchments together (rs=-0.37). For the correlations among catchment attributes themselves (Figure D 2), the highest value was between mean snow cover and NDVI (rs=-0.40), and between snow cover and fraction of steep area (rs=-0.40). this indeed reflects the relationship shown in **Figure 3**. between these three attribute groups behaving similarly. However, these results suggest that there are clear difficulties to derive correlations among attribute groups, or between attributes and signatures, without constraining the analysis to specific nested basins (e.g., Moselle)."*

*Appendix D: Extra scatter plots between attributes groups and baseflow index*

[Figure]

**Figure D 1: Scatter plots between the attributes with the most often highest correlations for each of the six groups (climatic, regional, continental and global geology, topography, soils and land use) and baseflow index. All 4,469 catchments are plotted in light blue, but the catchments of the five basins further explored are plotted with different colors according to the legend. Note that the Spearman correlation between each attribute and baseflow index is depicted in each panel.**

[Figure]

*Figure D 2: Scatter plots between the attributes with the most often highest correlations for each of the six groups (climatic, regional, continental and global geology, topography, soils and land use). All 4,469 catchments are plotted in light blue, but the catchments of the five basins further explored are plotted with different colors according to the legend.*

- Fig. 4: this comparison between the effects (or correlations) of global geology vs continental geology is useful, but it would benifit from shwong a but more than just the data points and 1:1 line. What are the average values of both (you can show the average X and Y coordiante, and this summarizes how well one predicts vs the other, and how well they overall predict). You can also show the correlation (coefficient) between the two data points to show how strongly related they are to another. Such things could greatly help the interpretation of these plots.

We thank the referee for this comment. These are extremely valuable insights into the figure. We took them into consideration and here you may find an updated version of it. Besides that, we have added some new information in the figure (different symbols for the higher attribute among geological for global), and added some new discussion and text in the manuscript as follows:

*L403-419: Across the four permeability classes in each geological map, the correlation with the six streamflow signatures varied notably (**Figure 4** and **Table 6**). For the baseflow index, the global map most frequently showed the strongest correlation in the high-permeability class (25 basins), followed by medium-low (15), low (13), and medium-high (10). In contrast, the continental map showed a different pattern: medium-low permeability had the highest number of leading correlations (19), followed closely by high (17), low (15), and medium-high (12).*

*This shift in the class with the highest $|r_s|$ between the two maps was evident for other streamflow signatures as well. For example, the continental map often highlighted the low-permeability class as having the strongest correlation —particularly for σHFD and σq_5—whereas the global map more frequently emphasized the high-permeability class, especially for σslope and σq_95.*

*Overall, the results suggest that the global map tends to emphasize the influence of high-permeability features, while the continental map captures a more nuanced influence of medium and low-permeability classes. This difference suggests that differences in geological map details can alter the inferred streamflow-geology relationships. Additional results for other attribute groups are provided in the supplementary material.*

**Table 6: Number of basins where each geological permeability attribute showed the strongest correlation ($|r_s|$) with each streamflow signature, for the global and continental geological maps.**

| Signature | Global map | | | | Continental map | | | |
|---|---|---|---|---|---|---|---|---|
| | High perm. | Medium-low perm. | Medium-high perm. | Low perm. | High perm. | Medium-low perm. | Medium-high perm. | Low perm. |
| σq_mean | 20 | 10 | 16 | 17 | 19 | 9 | 14 | 21 |
| σslope | 29 | 13 | 11 | 10 | 20 | 11 | 16 | 16 |
| σBFI | 25 | 15 | 10 | 13 | 17 | 19 | 12 | 15 |
| σHFD | 24 | 18 | 10 | 11 | 17 | 13 | 13 | 20 |
| σq_5 | 19 | 16 | 15 | 13 | 12 | 15 | 15 | 21 |
| σq_95 | 24 | 8 | 14 | 17 | 21 | 8 | 12 | 22 |

[Figure]

*Figure 4: Scatter plot of the |rs| derived from the global geology map, versus the |rs| from the continental geology map for the six streamflow signatures. Each light blue circle represents one of the 63 river basins. The 1:1 line is shown in dashed light gray, and the mean |rs| coordinates per geological map are depicted in black on both axes. Moreover, the subplots show the correlation between the correlations from both maps.*

---

## Author Response (AR2)

To facilitate the review process, all referee comments are in **black** while the authors response in **blue**.

**Final response**

Dear Authors,

Glad to see you have successfully addressed most comments by the two referees. However, one of them still has concerns with the statement, which is fully consistent with your results. Please check the comments and the statements/results carefully and revise the manuscript accordingly. I would be happy to make another round of review. Thanks.

Best,
Fuqiang
Tsinghua University

We thank the editor and reviewers for dedicating their time and expertise to review our manuscript. We have carefully addressed each point in detail and updated the manuscript accordingly. We hope that the current version of the manuscript is now ready for publication. The response is organized below addressing each of the points mentioned, with specific details regarding referee comments.

I would like to thank the authors for their careful efforts in revising the manuscript. One of the most significant improvements is the rephrasing to avoid overstated claims, which helps present the results more accurately. Nevertheless, I still find some statements that are not fully supported by the data, and I recommend a further round of moderate revision.

We thank the reviewer for the positive feedback. We have carefully addressed each point and implemented the corresponding changes in the manuscript. Additionally, we would like to highlight that we revised the terms "higher" and "lower" for the correlations to "stronger" and "weaker," following a suggestion from referee 2, to avoid confusion with the signs that $r_s$ may carry.

1. The authors have replaced evaluative terms such as "better" with more neutral language when presenting the results. However, some conclusions remain inadequately supported. For example, in the large-scale analysis paragraph of the Conclusions section, the authors state that "the continental map typically provided higher correlations on average…". While this is true for the mean |rs| (0.42 vs. 0.40), the difference is minimal, and adding or removing a single catchment could change the conclusion. Furthermore, Figure 3 does not clearly indicate in how many catchments the continental map yields higher rs; visually, it appears roughly balanced. I suggest the authors include quantitative metrics to clarify this point, such as the statistical significance of the difference or the proportion of basins in which the continental map provides higher rs.

We thank the reviewer for pointing out this issue in detail. The previous conclusion was indeed misleading relative to our main results. Global and continental maps provide complementary strengths, as we have already discussed earlier in the manuscript. Although the continental map presented consistently high correlations with Q95 and Q mean, for example, the results were more evenly distributed for most of the signatures (including baseflow index). Therefore, we have now carefully reviewed and adjusted the full manuscript to present the conclusions as fairly as possible. As part of these modifications, we also added the proportion of basins where the continental map provides higher correlations than the global (**Figure 4**; **L374–378**) and stressed that neither map consistently outperformed the other (e.g., **L20**, **L378–379**) across all signatures.

Particularly, here we have some of the modifications:

**L19-L20:** *"At this scale, continental and global geology maps produced different correlation patterns, with neither consistently superior."*

**L378–379:** *"This indicates that neither map consistently outperformed the other for all signatures at the same time at the large-scale analysis."*

**L523-527:** *"Our comparison of the two geological maps revealed markedly different correlation patterns with groundwater-oriented streamflow signatures, such as baseflow index and slope of the FDC (Figure 4). The absence of a consistently superior map suggests limited reliability when using broad-scale geological data for hydrological inference. This inconsistency likely reflects both uncertainties in the geological maps themselves and the difficulty of translating lithological information into hydrologically meaningful indicators, such as through reclassification into relative permeability classes."*

**L638-640:** *"our results showed limited consistency between them, and that neither map consistently outperformed the other in terms of correlation with groundwater-oriented streamflow signatures."*

2. Related to the first comment, I find Figure 3 insufficiently informative. Although the authors have added several Appendix figures as suggested by Reviewer #2, I believe the intent was to display scatterplots comparing rs values computed from different attribute groups. Specifically, the x-axis could represent rs calculated using geology attributes, and the y-axis those calculated using other attribute groups. Such plots would provide deeper insight.

We thank the reviewer for the valuable suggestion. After some discussion, we have decided to modify **Figure 3**, and to insert the suggested new illustration in the **Appendix (Figure D3)**. Now, **Figure 3** has its correlations

ordered from the attribute group with the highest number of basins to the lowest and have their respective number of basins within. In this way, we believe that readers can now extract useful information from the figure: how basins behaved differently in terms of correlations reached from each group. ranking of groups, the weak correspondence between most of the groups (which is also complemented by the new figure in **Appendix**). Finally, the figure also makes it clearer the inconsistence between the maximum $r_s$ among the geological maps, for instance.

[Figure]

*Figure 3. Maximum |rs| values for each catchment attribute group for each river basin. Each color represents the respective maximum |rs| value for a specific catchment attribute group (e.g., climate is shown in blue). The IDs of the Cinca (ES000331), Garonne (FR001604), Vienne (FR003986), Moselle (DEBU1959), and Narew (PL000936) basins are indicated in red. The groups are ordered in descending order starting with the group that ranked the most basins (land use) and ending with the least (continental geology). The number of basins is also indicated in the plot area.*

[Figure]

*Figure D3: Scatter plots between the different attribute groups for their maximum |rs| values for the baseflow index. Each light blue circle represents one of the river basins evaluated, with the five selected river basins highlighted using distinct colors: green (Cinca), pink (Garonne), orange (Vienne), red (Moselle) and blue (Narew). Moreover, the subplots show the correlation between the correlations from both maps.*

3. Regarding the small-scale analysis, I question whether a consistent correlation sign is truly an appropriate indicator. Runoff generation mechanisms differ among catchments, so the sign of the correlation may legitimately vary. For instance, t_flat_area_fra exhibits high |rs| values in all catchments except one, where the sign differs. This exception may not be problematic; rather, it could indicate a distinct runoff mechanism in the Sure subcatchment. I recommend that the authors interpret the correlation results more appropriately, acknowledging potential process differences.

We thank the reviewer for bringing up this discussion. We have carefully reviewed the text while taking this into consideration, making sure to clearly indicate when we argue that the sign consistency is related to our hypothesis (**L21-22; L657-659**), and to incorporate the proposed observation in the results section (**L502-507**). In particular, we have added the following sentences:

**L21-22:** *"The small-scale experiment reinforced these findings, as the regional map highlighted controls more consistent with process understanding."*

*L502-507: "It is important to highlight that high and sign consistent correlations do not necessarily imply causality even if they align with process understanding and expectations (i.e., hypothesis 1 and 2 in section 3.3). It is well known that dominant processes vary widely as well their controls. For instance, the fraction of flat area correlated strongly with σBFI in all sub-catchments, yet the Sure exhibited a positive sign, suggesting a different process dominance there. These results emphasize the importance of complementing statistical approaches with detailed hydrological insights to improve the understanding of hydrological controls."*

*L657-659: "However, the regional map was the only one that provided sign consistent and relatively high correlations across all sub-catchments, aligning with our initial, physically motivated hypotheses about the role of geology in baseflow variability in the Moselle basin."*

4. L658: This appears incorrect, as the global geology map does not yield the lowest rs for BFI (0.54, which is higher than the rs produced by climate and soil attributes).

We thank the reviewer for pointing out this mistake. We have corrected the phrase by removing the incorrect statement, which now reads as:

*L463-464: "Using the global attributes, geology ranked below at least one other landscape or climate attribute across all signatures."*

**Response to referee 2**

To facilitate the review process, all referee comments are in **black** while the authors response in **blue**.

I thank the authors for these revisions that successfully address all reviewer comments. One small note: To clarify the text further, I'd recommend using "stronger/weaker correlation" instead of "higher/lower correlation". This avoid ambiguity in cases of positve and negative correlations

We thank the reviewer for their positive feedback and suggestion. We have taken it into consideration throughout the text.